# Reasoning Multi-Agent Behavioral Topology for Interactive Autonomous Driving

Haochen Liu[1,2]  Li Chen[2,3]  Yu Qiao[2]  Chen Lv[1†]  Hongyang Li[2,3†]

[1] Nanyang Technological University  [2] Shanghai AI Lab  [3] University of Hong Kong

## Abstract

Autonomous driving system aims for safe and social-consistent driving through the behavioral integration among interactive agents. However, challenges remain due to multi-agent scene uncertainty and heterogeneous interaction. Current dense and sparse behavioral representations struggle with inefficiency and inconsistency in multi-agent modeling, leading to instability of collective behavioral patterns when integrating prediction and planning (IPP). To address this, we initiate a topological formation that serves as a compliant behavioral foreground to guide downstream trajectory generations. Specifically, we introduce **Behavioral Topology (BeTop)**, a pivotal topological formulation that explicitly represents the consensual behavioral pattern among multi-agent future. BeTop is derived from braid theory to distill compliant interactive topology from multi-agent future trajectories. A synergistic learning framework (BeTopNet) supervised by BeTop facilitates the consistency of behavior prediction and planning within the predicted topology priors. Through imitative contingency learning, BeTop also effectively manages behavioral uncertainty for prediction and planning. Extensive verification on large-scale real-world datasets, including nuPlan and WOMD, demonstrates that BeTop achieves state-of-the-art performance in both prediction and planning tasks. Further validations on the proposed interactive scenario benchmark showcase planning compliance in interactive cases. Code and model is available at https://github.com/OpenDriveLab/BeTop.

## 1   Introduction

Autonomous driving system aspires to safe, humanoid, and socially compatible maneuvers [1]. This drives for formulation, prediction, and negotiation of collective future behaviors among interactive agents and autonomous vehicles (AVs) [2]. Remarkable accuracy is achieved by learning-based paradigms [3], including end-to-end modular design [4–7], social modeling [8, 9], and trajectory-level integration [10–13]. However, substantial challenges arise in real-world cases due to scene uncertainty and volatile interactive patterns for multi-agent future behaviors.

To embrace compliant patterns for multi-agent future behaviors, current formulations fall into two mainstreams, dense representation and sparse representation (Fig. 1). Dense representation quantizes agent behaviors under ego-centric rasterization, forecasting bird's eye view (BEV) occupancy probabilities [14, 7, 15] or temporal flow [16–18]. It is easy to deduce interactions, perform scalable behaviors for agents [19], and align with BEV perceptions [20]. Still, dense representation is hindered by frozen receptions. It causes safety-vulnerable intractability and occlusions potentially interacting with ego maneuvers [16, 21]. Contrary to pixel-wise behavioral probability, sparse representation forecasts agent-anchored set of trajectories [22–25] or intention distributions [10, 26, 27]. Its multimodal formulation for each agent marks the elasticity in diverse behavioral uncertainty and tractability under flexile spatial semantics. However, behavioral misalignment [28] and modality collapse [24]

---

Work done while Haochen's internship at Shanghai AI Lab. [†] Equal co-advising.

38th Conference on Neural Information Processing Systems (NeurIPS 2024).

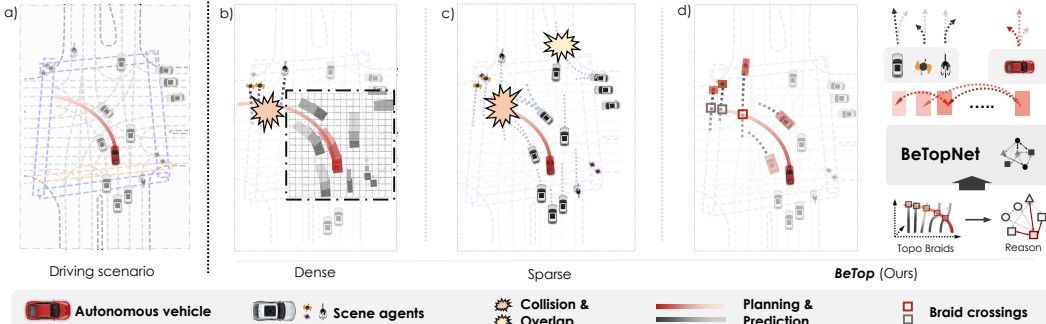

Figure 1: **Multi-agent Behavioral Formulation. (a)** A typical driving scenario in Arizona, US [35]; **(b)** Dense representation conducts scalable occupancy prediction jointly, but restrained reception leads to unbounded collisions with planning; **(c)** Sparse supervision derives multi-agent trajectories with multi-modalities, while it struggles with conflicts among integrated prediction and planning; **(d)** BeTop reasons future topological behaviors for all scene-agents through braids theory, funneling interactive eventual agents (in highlighted colors) and guiding compliant joint prediction and planning.

impede compliant multi-agent modeling, requiring exponential computations with growth agent numbers [29]. Issues particularly result in unstable and slow behavioral learning when exposed to predictions and planning (IPP) [30]. Typical solutions by conditional prediction [31, 28, 32] or game-theoretic reasoning [8, 33] often lead to nonstrategic maneuvers [34] due to non-compliant rollouts in adjusting interactive behaviors. This calls for a re-formulation for multi-agent behaviors, which should stabilize collective behavioral patterns in a compliant manner for IPP objectives.

The decision-making process for human drivers provides valuable insights. Humans primarily determine the future behavior of interacted agents for decision-making without relying on their specific states [2, 36]. Thus, an effective strategy involves assessing agent-wise behavioral impact on planning maneuvers, and reasoning about compliant interactions. Our fundamental insight is that compliant multi-agent behaviors exhibit topological formations, which can be identified by distilling consensual interactions from future behaviors. Prior works have approached this challenge through structural design [37, 38] or implicit relational learning [39–41] using GNNs [42] or Transformers [43]. Other studies quantify uncertainty by topological properties [44, 45]. Nevertheless, current literature is scarce in formulating explicit future supervision of compliant multi-agent behavioral patterns.

To this end, we launch the multi-agent behavior formulation termed as ***Behavioral Topology*** **(BeTop)**. At its core, BeTop explicitly forms the topological supervision of consensual multi-agent future interactions, and reasons to guide prediction and planning. BeTop stems from braid theory [46], which infers compliant interactions of multiple paths from the intertwining of their braids. This empowers BeTop to intuitively distill forward intertwines (occupancy) as joint topology from braided multi-agent future trajectories (Fig. 1), marrying dense and sparse representation. With the aid of BeTop, we introduce a synergistic Transformer-based learning stack, BeTopNet, for learning IPP objectives. To implement, an iterative decoding strategy simultaneously reasons about Behavioral Topology and generates trajectory sets. Then the topology-guided local attention, embedded in each decoder layer, selectively queries behavioral semantics from social-compliant agents within the predicted BeTop priors. To further alleviate multi-agent uncertainty through topological guidance, a contingency planning paradigm is fitfully deployed. We lay out the imitative contingency learning process, which regulates the safety-ensured short-term plan. It maintains the long-range uncertainty by reasoned joint predictions from BeTop. Experimental results exhibit enhanced consistency and accuracy for prediction and planning in real-world scenarios. Testing in proposed interactive cases further highlights the planning ability of BeTopNet. To sum up, our contributions are three-fold:

- We bring in the concept of Behavioral Topology, a multi-agent behavioral formulation for topological reasoning that explicitly supervises consensual future interactions jointly for the IPP system.

- A synergistic learning framework BeTopNet, offering joint planning and prediction guided by topology reasoning, is devised. Topology-guided local attention and imitative contingency planning could resolve scene compliance and multi-agent uncertainty.

- Benchmarking on nuPlan [47] and WOMD [35], our approach demonstrates strong performance in both planning strategy and prediction accuracy. BeTopNet witnesses evident improvement

over previous counterparts, *e.g.*, $+7.9\%$ in general planning score, $+3.8\%$ under interactive cases, $+4.1\%$ mAP for joint prediction, and $+2.3\%$ mAP for marginal prediction.

## 2 Related work

**Multi-agent behavioral modeling.** Carving the collective future behavior of diverse agents is imperative for socially-consistent driving maneuver. Earlier approaches centered around occupancy prediction [14, 5, 4, 48]. Forecasting the spatial presence under dense BEV representation [49, 20] offers flexibility of arbitrary agents [17, 50, 19] and alignment with perception [7, 51]. However, rigidity in resolution induces scenario occlusion [16], rendering intractable occupancy [21]. In parallel, sparse representation consolidates multi-agent behavior into cohesive modalities across future trajectories [52, 22, 53, 54] or intentions [27, 10, 26, 55]. Joint future behaviors are derived through goal-based sampling or recombination from marginal predictions [56–58]. However, the collection of joint modalities is susceptible to mode collapse and entails exponential complexity [29, 59]. Meanwhile, topological representation has garnered traction as motion primitives [60, 61] or tools [44, 45] for scenario quantification, yet topological properties delineating collective future behaviors remain largely unexplored. Our BeTop targets the issue, marrying dense behavior probabilities by topology with the sparse motion from joint predictions to present structured future behaviors.

On the other hand, inconsistent communal agent behaviors have motivated leveraging future interactions. Implicit approaches obtain tacit interactions with attention [62, 32, 63] or GNN [24, 64, 37, 65] from final motion regressions. Nonetheless, the implicit supervisions are found inefficient in dynamic scenarios [22]. Contrarily, explicitly reasoning mutual behaviors by conditional factorization [31, 66], relation reasoning [41, 40], or entropy-based methods [67, 68] offer consistent behavioral priors. However, hefty variance across agent dynamics and scenario geometries yield unstable inference. Distinguished from them, BeTop crafts a compact topological supervision that stabilize future interactions among multi-agent behaviors. Derived from topological braids, BeTop offers a topological equivalent behavioral representation to guide compliant forecasting and planning.

**Integrated prediction and planning.** IPP system aims to harmonize trajectory-based learning of future interactive behaviors between the ego vehicle and social agents. Rule-based approaches [69–71] integrate handcrafted future interactions to evaluate candidate planning profiles, offering remarkable outcomes in rule-powered reactive simulation [47]. Still, the absence of real-world behaviors exhibits significant gaps in interactive scenarios. Learning-based methods yield imitative planning by integrating predictions within holistic modeling [72–74]. However, history-based coalitions pose challenges in supervising future homology among agents. Recently, hybrid pipelines [8, 75] have utilized post-processing and optimization upon learning-based models to realize behavior interactions among predictions and planning. This can entail significant computational overhead, and imitative planning tends to overestimate hereditary uncertainty in behavior predictions. Tree-based [10, 27] and contingency-enabled [76, 11] works seek to balance planning preemption and aggression in the face of behavior uncertainty. Nonetheless, pipelines without holistic interactions fall into passive planning maneuvers and incur high exponential cost for predictions. In our work, BeTop provides an explicit prior for future interactive behaviors which enhances compliant trajectory generations. Moreover, the synergistic prediction and contingency planning networks with BeTop effectively manage behavioral uncertainty.

## 3 Behavioral Topology

Presenting BeTop, we commence the Behavioral Topology formulation and task statement of IPP for autonomous driving in Sec. 3.1. Then, we demonstrate the BeTopNet network architecture (Fig. 3) for topological reasoning and IPP generation in Sec. 3.2. Finally, in Sec. 3.3, we propose the imitative contingency learning process by topological guidance for the proposed network.

### 3.1 Formulation

**Problem formulation.** We consider the driving scenario with $N_a$ agents as $A_{1:N_a}$ at presence $t = 0$, along with the scenario map $\mathbf{M}$. The states over historical horizon $T_h$ are denoted as $\mathbf{X}_1$ for AV and as $\mathbf{X}_{2:N_a}$ for scenario agents, respectively, where $\mathbf{X}_n = \{\mathbf{x}^{-T_h:0}\}_n, n \in [1, N_a]$. The objective for integrated prediction and planning is to jointly predict scene agents' trajectories $\mathbf{Y}_{2:N_a}$ as well as AV planning $\mathbf{Y}_1$ over a future horizon $T_f$ as $\mathbf{Y}_n = \{\mathbf{y}^{1:T_f}\}_n, n \in [1, N_a]$.

Figure 2: **BeTop formulation.** Joint future trajectories are transformed to braid sets, and then form joint topology through intertwine indicators.

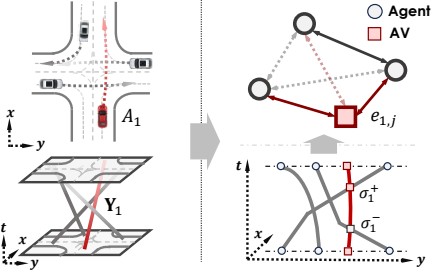

Table 1: **Analysis on different behavioral formulations.** BeTop labels behave most similarly to human annotations [35], excelling over other formulations like $k$ nearest GT or local attention.

| Behavioral | WOMD | |
| Formulations | Acc. ↑ | AUC ↑ |
|---|---|---|
| *Expert* [35] | 1.000 | 1.000 |
| GT top-$k$ | 0.833 | 0.702 |
| Local attention [22] | 0.951 | 0.522 |
| JFP graph [67] | 0.955 | 0.500 |
| **BeTop (Ours)** | **0.967** | **0.731** |

**Topological formulation.** We leverage the braid theory [46], which probes explicit formulations for compliant multi-agent interactions from future data $\mathbf{Y}_{1:N_a}$. Intuitively, it denotes a transform process for $\mathbf{Y}_{1:N_a}$ with respective agent coordinates, and then gathers each future forward intertwine (occupancy) as joint interactions. Formally, consider the braid group $\mathbf{B}_{N_a} = \{\sigma_n\}$ by $N_a$ primitive braids $\sigma_n$, each of which $\sigma_n = (f_1^n, \cdots, f_{N_a}^n)$ denotes a tuple of monotonically increased functions $f : \mathbb{R}^3 \times \mathbf{Y} \to \mathbb{R}^2 \times I$ mapping from Cartesian $(\vec{x}, \vec{y}, \vec{t})$ to lateral coordinate $(\vec{y}, \vec{t})$ for agent future $\mathbf{Y}$. Specifically, the function $f_i^n$ in $\sigma_n$ is defined as $f_i^n \to (\mathbf{Y}_i - \mathbf{b}_n)\mathbf{R}_n; 1 \le i, n \le N_a$, where $\mathbf{b}_n$ and $\mathbf{R}_n$ denote the left-hand transform matrix to local coordinate of agent $A_n$. The joint interactive behaviors are identified as a set of braids having intertwines $\{\sigma_n^{\pm}\} \subset \mathbf{B}_{N_a}$ over others [45], as shown in Fig. 2. Opposite to implicit methods [22, 67, 41] banking on future distance heuristic, each intertwine in the braid can signify an explicit behavioral response, distinguishing between assertive ($\sigma_n^+$, elicit yielding from others) and passive ($\sigma_n^-$, yield to others) maneuvers. To avert difficulties in dynamic braid set inference, we redraft multi-agent braids from a topology reasoning perspective.

Named by BeTop, the goal is to reason a topological graph $\mathcal{G} = (\mathcal{V}, \mathcal{E})$ for multi-agent future behaviors (Fig. 2). Expressly, node topology $\mathcal{V} = \{\mathbf{Y}_n\}$ is denoted by multi-agent future trajectories. We can then reformulate the braid set $\{\sigma_n^{\pm}\}$ as an edge topology $e_{ij} \to \mathcal{E} \in \mathbb{R}^{N_a \times N_a}; 1 \le i, j \le N_a$ for future interactive behaviors. Each topology element $e_{ij}$ can be defined by two braid functions $f_i^i, f_j^i \in \sigma_i$ assessing the future intertwines along with $\mathbf{Y}_i, \mathbf{Y}_j$ as: $e_{ij} = \max_t \mathbf{I}\left(f_i^i(\mathbf{y}_i^t), f_j^i(\mathbf{y}_j^t)\right)$. Here $\mathbf{I}$ is an intertwine indicator by segment intersection [77] under lateral coordinates. With favorable properties proved in Appendix B, we can formulate the reasoning task as:

$$\mathcal{G}^* = (\max \hat{\mathcal{V}}, \max \hat{\mathcal{E}}). \tag{1}$$

Agent future $\hat{\mathbf{Y}}$ in node term $\hat{\mathcal{V}}$ is defined by Gaussian mixtures (GMM) and optimized in Sec. 3.3. The edge topology reasoning $\hat{\mathcal{E}}$ can be specified as a probabilistic inference problem by:

$$\max \hat{\mathcal{E}} = \max \sum_i \sum_j e_{ij} \log \hat{e_{ij}} + (1 - e_{ij}) \log(1 - \hat{e_{ij}}), \tag{2}$$

where $1 \le i, j \le N_a$. Synergistic reasoning structures are then established optimizing $\mathcal{G}^*$.

**Comparative analysis.** To highlight BeTop's position among various formulations, we first conduct a preliminary analysis to assess behavioral similarity by retrieving future interactive agent pairs using human annotations [35]. Human likeness is quantified by classification metrics, including accuracy and the area under the curve (AUC), with annotated interactive IDs. As depicted in Table 1, labeled BeTop achieves the closest behavioral similarity compared with other well-accepted formulations in the community. Compared with retrieving $k$ nearest strategy ($k = 6$) by ground-truth future states, we observe advanced differentiation in non-interactive behaviors ($+16.1\%$ Acc., $+4.13\%$ AUC) by BeTop. We then look into the generic learning-based structure by attention [22] or dynamic graph [67] for interactive behaviors. Despite high accuracy, their inferior AUC scores imply difficulties in retrieving precise interactivity compared with BeTop ($+19.9$ AUC). We refer analytical content in Appendix B. This draft for a reasoning framework BeTop prompting joint behaviors.

### 3.2    BeTopNet

As presented in Fig. 3, we introduce the synergistic learning framework reasoning BeTop in response to the series of challenges. It encompasses a Transformer backended encoder-decoder network. With

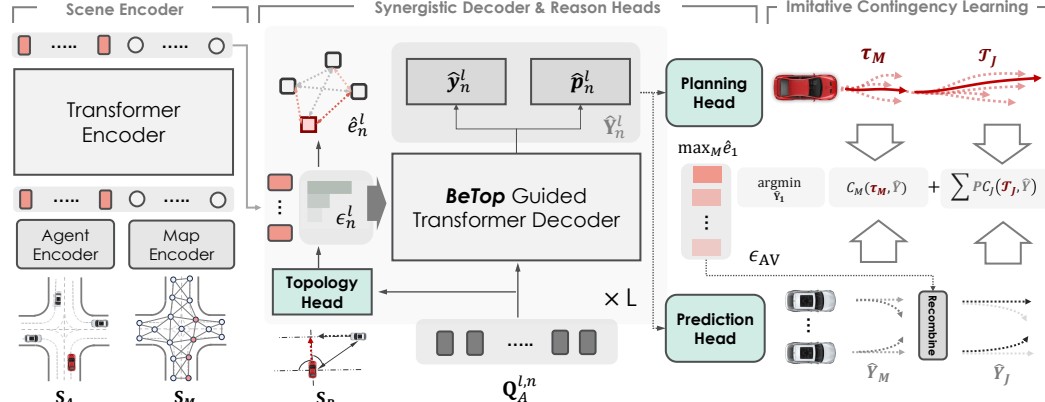

Figure 3: **The BeTopNet Architecture.** BeTop establishes an integrated network for topological behavior reasoning, comprising three fundamentals. Scene encoder generates scene-aware attributes for agent $\mathbf{S}_A$ and map $\mathbf{S}_M$. Initialized by $\mathbf{S}_R$ and $\mathbf{Q}_A$, synergistic decoder reasons edge topology $\hat{e}_n^l$ and trajectories $\hat{\mathbf{Y}}_n^l$ iteratively from topology-guided local attention. Branched planning $\tau \in \hat{\mathbf{Y}}_1$ with predictions and topology are optimized jointly by imitative contingency learning.

encoded scene semantics $\mathbf{X}; \mathbf{M}$, the proposed network features a synergistic decoder which reasons and guides BeTop. Reason heads for topology $\hat{\mathcal{E}}$ and IPP for $\hat{\mathcal{V}}$ comprise the behavioral graph $\mathcal{G}$.

**Scene encoder.** We leverage a scene-centric coordinate system following planning-oriented principle [7]. Scene attributes comprise historical agent states $\mathbf{X} \in \mathbb{R}^{N_a \times T_h \times D_a}$ and map polyline inputs $\mathbf{M} \in \mathbb{R}^{N_m \times L_m \times D_m}$, where we portion $N_m$ map segments with length $L_m$ from full scene map. Both attributes are encoded separately as $\mathbf{S}_A \in \mathbb{R}^{N_a \times D}$ and $\mathbf{S}_M \in \mathbb{R}^{N_m \times D}$ and concatenated as scene features $\mathbf{S} = [\mathbf{S}_A; \mathbf{S}_M] \in \mathbb{R}^{(N_a + N_m) \times D}$. A stack of Transformer encoders with local attention are directly employed in capturing regional interactions from encoded scene semantics $\mathbf{S}_A, \mathbf{S}_M$.

**Synergistic decoder.** Retaining encoded scene features $\mathbf{S}_A, \mathbf{S}_M$, we zoom in the decoding strategy that asks for: 1) interactively reason simultaneous BeTop formulations; 2) selectively decoding of compliant interactive semantics leveraging reasoned topology priors. To this end, we introduce the iterative process of $N$ Transformer decoder layers contributed to all agents, pursuing the basis from [78]. To iron out the scene uncertainties, a multi-modal set of $M$ decoding queries $\mathbf{Q}_A^0 \in \mathbb{R}^{M \times D}$ are initialized for multi-agent future trajectories. Meanwhile, relative attributes $\mathbf{S}_R \in \mathbb{R}^{N_a \times N_a \times D_R}$ are deployed through MLPs as topology features $\mathbf{Q}_R^0 \in \mathbb{R}^{N_a \times N_a \times D}$ for edge topology reasoning.

Next, we devise dual infostreams to the iterative decoding process for $\hat{\mathcal{V}}$ of future trajectories and $\hat{\mathcal{E}}$ of future topology. Given agent $A_n$, the decoding process in layer $l$ follows:

$$\mathbf{Q}_R^{l,n} = \text{TopoDecoder}\left(\mathbf{Q}_A^{l-1,n}, \mathbf{Q}_R^{l-1,n}, \mathbf{S}_A\right), \hat{e}_n^l = \text{TopoHead}\left(\mathbf{Q}_R^{l,n}\right), \quad (3)$$

$$\mathbf{Q}_A^{l,n} = \text{TransDecoder}\left(\mathbf{Q}_A^{l-1,n}, \mathbf{S}_A, \mathbf{S}_M, \hat{\mathbf{Y}}_n^{l-1}, \hat{e}_n^l\right), \hat{\mathbf{Y}}_n^l = \text{IPPHead}(\mathbf{Q}_A^{l,n}), \quad (4)$$

where both future trajectories $\hat{\mathbf{Y}}_n \in \hat{\mathcal{V}}$ and interactive topology $\hat{e}_n \in \hat{\mathcal{E}}$ in BeTop are decoded in synergistic manners. Reasoned edge topology $\hat{e}_n^l \in \mathbb{R}^{M \times N_a}$ are garnered by topological decoder with query broadcasting $\mathbf{Q}_A^{l-1,n}$; Reasoning nodes for $\hat{\mathbf{Y}}_n$, a Transformer decoder with topology-guided local attention are drafted serving $\hat{e}_n^l$ as priors. We provide further details in Appendix C.1.

**Topology-guided local attention.** Querying whole-scene agent semantics results in misaligned interactive agents and sparse attention. This motivates our design for local attention guided by the reasoned topology $\hat{e}_n^l \in \mathbb{R}^{M \times N_a}$ as priors. Specifically, we retrieve the top-$K$ index $\epsilon_n^l \in \mathbb{R}^{M \times K}$ priored from $\hat{e}_n^l$ for eventual interactive agents behaviors with $A_n$. Interactive indices are directly leveraged in gathering $\mathbf{S}_A$ selectively for local cross-attention. This process is formed as:

$$\mathbf{C}_A^{l,n} = \text{TopoAttn}\left(\mathbf{Q}_A^{l-1,n}, \mathbf{S}_A, \hat{e}_n^l\right) \rightarrow \text{MultiHeadAttn}\left(q = \mathbf{Q}_A^{l-1,n}; k, v = \mathbf{S}_A^{i \in \epsilon_n^l}\right), \quad (5)$$

where $\epsilon_n^l = \text{argmax}_K(\hat{e}_n^l)$. Topology-guided agent features $\mathbf{C}_A^{l,n}$ are then aggregated in each layer.

**Reason heads.** Given respective decoding features $\mathbf{Q}_R^{l,n}$ and $\mathbf{Q}_A^{l,n}$ for each layer, we affix reason heads accustomed to corresponding formulations for $\hat{e}_n$ and $\hat{\mathbf{Y}}_n$. Referred in Eq. (3), the topology head, planning head, and prediction head (IPP heads) are jointly devised by stacked MLPs in reasoning BeTop results. For agent $A_n$ in each layer, reason heads decode GMM components of future states $\hat{\mathbf{y}}_n \in \mathbb{R}^{M \times T_f \times 5}$ (referring to $(\mu_x, \mu_y, \log \sigma_x, \log \sigma_y, \rho)$ per step) with mixture score $\hat{\mathbf{p}}_n \in \mathbb{R}^M, \{\hat{\mathbf{y}}_n, \hat{\mathbf{p}}_n\} \in \hat{\mathbf{Y}}_n$, as well as interactive edge topology $\hat{e}_n^l \in \mathbb{R}^{M \times N_a}$ for BeTop.

### 3.3 Imitative Contingency Learning

Pursuing the target in Eq. (1), BeTopNet learns end-to-end objectives imitating human-like multi-agent behaviors, integrating compliant behaviors by contingency planning under scenario uncertainties.

**Imitation learning.** Imitation objectives are firstly established in regulating multi-agent behavioral states $\{\hat{\mathbf{Y}}_n\} \subset \hat{\mathcal{V}}$ while maximizing their interactive distributions $\hat{\mathcal{E}}$. The imitative objective for $\hat{\mathbf{Y}}$ is defined by the negative log-likelihood (NLL) from best-reasoned components $m^*$ closest to ground-truths, as denoted: $\mathcal{L}_\mathcal{V} = \sum_t^{T_f} \mathcal{L}_{\text{NLL}}(\hat{\mathbf{y}}_n^{m^*,t}, \hat{\mathbf{p}}_n^{m^*}, \mathbf{Y}_n)$. Followed Eq. (2), the behavioral distributions for edge topology are computed by binary cross-entropy (BCE) given gathered $\hat{e}_n^{m^*} \in \mathbb{R}^{N_a}$, formulated as $\mathcal{L}_\mathcal{E} = \sum_j^{N_a} \mathcal{H}(\hat{e}_{n,j}^{m^*}, e_{n,j})$ over $N_a$ agents jointly.

**Integrated contingency planning.** To integrate compliant behavior learning for $\mathcal{G}$ amidst multi-agent scenario uncertainties, contingency planning [79, 76] is turned out an apt solution. Bridging immediate safe maneuvers $\tau_M$ to branched planning sets $\{\tau_J\}$ with joint prediction, it adjourns uncertain decisions and ensures actual safety. While direct joint prediction may lose diversity [11], reasoned topology $\hat{\mathcal{E}}$ serves as a suitable medium distilling future interactive agents for efficient joint combination. Given imitative AV planning outputs $\tau \subset \hat{\mathbf{Y}}_1$ with branching time $t_b \in (1, T_f)$, integrating contingency learning asks for a safe short-term plan $\tau_M \in \mathcal{T}_M, \mathcal{T}_M \in \mathbb{R}^{M \times t_b \times 2}$ to full marginal predictions $\hat{Y}_M = \hat{\mathbf{Y}}_{2:N_a}$, as well as $M$ branched planning sets $\mathcal{T}_J^m = \{\tau_J^{1:M_b}\}_m$ guided by joint predictions $\hat{Y}_J^m$. This is defined by:

$$\tau_M^* = \underset{\tau \subset \hat{\mathbf{Y}}_1}{\operatorname{argmin}} \max_{\hat{Y}} C_M\left(\tau_M, \hat{Y}_M\right) + \sum_m P(\hat{Y}_J^m) C_J\left(\mathcal{T}_J^m, \hat{Y}_J^m\right), \tag{6}$$

where $\max_{\hat{Y}} C_M$ denotes worst-case cost fir $\tau_M$; Joint predictions $\hat{Y}_J$ with scene probabilities $P(\hat{Y}_J)$ are recombined by $K_M$ interactive agent subsets, indexing $\epsilon_{\text{AV}} \in \mathbb{R}^{K_M}$ from sorted AV topology: $\epsilon_{\text{AV}} = \operatorname{argmax}_{K_M}(\max_M \hat{e}_1)$. It is described by joint costs $C_J$ in guiding branched planning maneuvers. Specifically, both cost functions are defined by the repulsive potential field [8] discouraging planning proximity with respective prediction formulations.

**Training loss.** BeTopNet is trained end-to-end through imitative objectives and contingency planning costs by weighted integration for each layer, whenever applicable (for the datasets). Please refer to Appendix C.2 for additional details.

## 4 Experiment

With preliminary analysis in Sec. 3.1, this section further discovers the following questions: **1)** Can BeTop perform compliant planning via BeTopNet, especially in interactive scenarios? **2)** Can BeTop achieve accurate marginal and joint predictions of heterogeneous agents under diverse real-world cases? **3)** Can the formulated BeTop facilitate existing state-of-the-art prediction and planning methods? and **4)** How do the functionalities in BeTopNet affect the performance?

**Benchmark and metrics.** BeTop is verified on diverse benchmarks. We leverage two large-scale real-world datasets, *i.e.*, nuPlan [47] and Waymo Open Motion Dataset (WOMD) [35], which are presently the most diverse motion datasets in manifesting planning and prediction performance. For planning tasks in nuPlan, there are in total 1M training cases with 8s horizons. 8,300 separated testing set are chosen by *Test14-Hard* and *Test14-Random* benchmarks [73] for hard-core and general driving scenes. With further demands verifying maneuvers under interactive cases, we build the *Test14-Inter* benchmark filtering 1,340 scenes by testing set. Scenarios ranging 15 seconds are tested under three tasks: 1) open-loop (OL), 2) close-loop non-reactive (CL-NR) simulations, and 3) reactive

Table 2: **Performance comparison of open- and closed-loop planning on nuPlan benchmarks.** BeTopNet positions top average planning score and non-reactive simulation amongst SOTA planning systems by all types (rule, learning, and hybrid), especially under difficult benchmarked scenarios.

| Type | Method | Test14 Hard | | | | Test14 Random | | | |
|---|---|---|---|---|---|---|---|---|---|
| | | OLS ↑ | CLS-NR ↑ | CLS ↑ | **Avg. ↑** | OLS ↑ | CLS-NR ↑ | CLS ↑ | **Avg. ↑** |
| *Expert* | *Log Replay* | 1.000 | 0.860 | 0.688 | 0.849 | 1.000 | 0.940 | 0.759 | 0.900 |
| Rule | IDM [70] | 0.201 | 0.562 | 0.623 | 0.462 | 0.342 | 0.704 | 0.724 | 0.590 |
| | PDM-Closed [69] | 0.264 | 0.651 | 0.752 | 0.556 | 0.463 | 0.901 | 0.916 | 0.760 |
| Hybrid | GameFormer [8] | 0.753 | 0.666 | 0.688 | 0.702 | 0.794 | 0.808 | 0.793 | 0.798 |
| | PDM-Hybrid [69] | 0.738 | 0.660 | 0.758 | 0.719 | 0.822 | 0.902 | 0.916 | 0.880 |
| Learning | UrbanDriver [74] | 0.769 | 0.515 | 0.491 | 0.592 | 0.824 | 0.633 | 0.610 | 0.689 |
| | PDM-Open [69] | 0.791 | 0.335 | 0.358 | 0.495 | 0.841 | 0.528 | 0.572 | 0.647 |
| | PlanCNN [72] | 0.524 | 0.494 | 0.522 | 0.513 | 0.629 | 0.697 | 0.675 | 0.667 |
| | GC-PGP [83] | 0.738 | 0.432 | 0.396 | 0.522 | 0.773 | 0.560 | 0.514 | 0.616 |
| | PlanTF [73] | 0.833 | 0.726 | 0.617 | 0.725 | 0.871 | 0.865 | 0.806 | 0.847 |
| | **BeTopNet (Ours)** | **0.840** | **0.771** | **0.688** | **0.766** | **0.876** | **0.902** | **0.857** | **0.878** |

Table 3: **nuPlan closed-loop planning results on the proposed interactive benchmark.** BeTopNet achieves desirable PDMScore, with planning safety, road compliance, and driving progress.

| Type | Method | Test14 Inter | | | | | | |
|---|---|---|---|---|---|---|---|---|
| | | Col. Avoid ↑ | Drivable ↑ | Direction ↑ | Progress ↑ | TTC ↑ | Comfort ↑ | **PDMScore ↑** |
| *Expert* | *Log Replay* | 1.000 | 1.000 | 1.000 | 0.881 | 1.000 | 0.999 | 0.950 |
| Rule | PDM-Closed [69] | 0.886 | 1.000 | 1.000 | 0.818 | 0.853 | 0.999 | 0.833 |
| Learning | Constant Acc. | 0.449 | 0.509 | 0.651 | 0.048 | 0.419 | **1.000** | 0.108 |
| | UrbanDriver [74] | 0.970 | 0.955 | 0.992 | 0.798 | 0.932 | **1.000** | 0.854 |
| | PlanCNN [72] | 0.902 | 0.895 | 0.973 | 0.678 | 0.859 | 0.999 | 0.720 |
| | PlanTF [73] | 0.982 | 0.946 | 0.992 | 0.825 | **0.952** | 0.999 | 0.871 |
| | **BeTopNet (Ours)** | **0.983** | 0.960 | 0.999 | 0.859 | 0.950 | 0.999 | **0.894** |

(CL-R) ones by nuPlan simulator. We report the official Planning Scores [80] computed by each task. The motion prediction tasks in WOMD share 487k training scenarios, with 44k validation and 44k testing set separately partitioned under two challenges: 1) The *Marginal prediction challenge* [81] forecasting multiple scene agents independently; 2) The *Joint prediction challenge* [82] predicting joint trajectory collections by two interactive agents. Primary metrics of mAP and Soft mAP are ranked for official leaderboards [81, 82]. We leave experimental details in Appendix D.

## 4.1 Main Result

**Performance for interactive planning.** Table 2 demonstrates the planning results under difficult and regular test cases. Notably, BeTopNet marks top average planning scores, achieving +7.9% in hard cases and excels +6.2% (CLS-NR) in closed-loop simulations. Specifically, it gains solid improvements against learning-based planners. This can be attributed to topological formulations learning stabilized joint behavioral patterns, boosting +6.2%, +4.3% non-reactive simulations by real-world logs and enhancing reactive simulation (+11.5%, +6.3%). Contingency objectives enhance uncertainty compliance, leading to expanded results in hard scenarios. Meanwhile, BeTopNet also outperforms rule-based and hybrid planning agents asking for post-optimizations [8] or hefty rules in coinciding with reactive simulation setups [69, 70]. We report +15.8% and +18.4% results of non-reactive simulation in hard cases and close performance in general scenes. Moreover, interactive planning compliance is also verified in the proposed *Test14-Inter* benchmark centering on interactive scenarios. As in Table 3, BeTopNet fosters +3.8% planning score over previous methods, marking +5.5% driving progress and +2.9% driving compliance closest to human performance. Qualitative results of interactive scenarios in Fig. 5(a-d) further corroborate planning compliance by BeTop.

**Performance for marginal and joint motion prediction.** Marginal prediction results are in Table 4. Without the aids of model ensembles or extra data [25, 59], BeTopNet outperforms existing approaches, manifesting +2.7% and +3.4% mAP metrics comparing concurrent methods [53, 86] for compliant predictions. Exhibited strong prediction displacement metric (−4.3% minFDE) over methods using extra pretraining [85], it should be noted that displacement metric is less illustrative as it discounts uncertainty scoring. BeTopNet further outperforms +6.0% and +26.1% Soft mAP over multi-agent predictors solely leveraging scenario attention [25] or graph [24]. N. Table 5 exhibits the

Table 4: **Performance of marginal prediction on WOMD Motion Leaderboard.** BeTopNet surpasses existing motion predictors without model ensemble or using extra data. [†] extra LIDAR data and pretrained model. Primary metric.

| Set split | Method | minADE ↓ | minFDE ↓ | Miss Rate ↓ | mAP ↑ | **Soft mAP ↑** |
|---|---|---|---|---|---|---|
| Test | ReCoAt [84] | 0.7703 | 1.6668 | 0.2437 | 0.2711 | - |
| | HDGT [24] | 0.5933 | 1.2055 | 0.1854 | 0.3577 | 0.3709 |
| | MTR [22] | 0.6050 | 1.2207 | 0.1351 | 0.4129 | 0.4216 |
| | MTR++ [25] | 0.5906 | 1.1939 | 0.1298 | 0.4329 | 0.4414 |
| | MGTR[†] [85] | 0.5918 | 1.2135 | 0.1298 | 0.4505 | 0.4599 |
| | EDA [53] | **0.5718** | 1.1702 | **0.1169** | 0.4487 | 0.4596 |
| | ControlMTR [86] | 0.5897 | 1.1916 | 0.1282 | 0.4414 | 0.4572 |
| | **BeTopNet (Ours)** | 0.5723 | **1.1668** | 0.1176 | **0.4566** | **0.4678** |
| Val | MTR [22] | 0.6046 | 1.2251 | 0.1366 | 0.4129 | - |
| | EDA [53] | **0.5708** | 1.1730 | 0.1178 | 0.4353 | - |
| | **BeTopNet (Ours)** | 0.5716 | **1.1640** | **0.1177** | **0.4416** | - |

Table 5: **Performance of joint prediction on WOMD Interaction Leaderboard.** BeTopNet outperforms in both mAP metrics. Primary metric.

| Set split | Method | minADE ↓ | minFDE ↓ | Miss Rate ↓ | **mAP ↑** | Soft mAP ↑ |
|---|---|---|---|---|---|---|
| Test | HeatIRm4 [37] | 1.4197 | 3.2595 | 0.7224 | 0.0804 | - |
| | M2I [31] | 1.3506 | 2.8325 | 0.5538 | 0.1239 | - |
| | GameFormer [8] | 0.9721 | 2.2146 | 0.4933 | 0.1923 | 0.1982 |
| | AMP [32] | 0.9073 | 2.0415 | 0.4212 | 0.2294 | 0.2365 |
| | MTR++ [25] | **0.8795** | **1.9505** | 0.4143 | 0.2326 | 0.2368 |
| | **BeTopNet (Ours)** | 0.9744 | 2.2744 | 0.4355 | **0.2412** | **0.2466** |
| Val | MTR [22] | 0.9132 | 2.0536 | 0.4372 | 0.1992 | - |
| | AMP [32] | **0.8910** | **2.0133** | 0.4172 | 0.2344 | - |
| | **BeTopNet (Ours)** | 0.9304 | 2.1340 | **0.4154** | **0.2366** | - |

joint prediction results. BeTopNet outperforms all methods in both mAP metrics ($+4.1\%, +3.7\%$ Soft mAP and mAP), presenting robust prediction compliance credit to BeTop formulations for stable future interaction patterns and aligned by local attention in BeTopNet. Particularly, BeTop shows interactive compliance, improving $+5.1\%$ mAP over recent auto-regressive approaches [32], boosting $+25.4\%$ mAP with game-theoretic methods [8] by a large margin. Fig. 5 (e-h) demonstrates the qualitative prediction performance by BeTopNet. At the time of submission, BeTopNet ranked $1^{st}$ on both WOMD prediction leaderboards [82, 81].

## 4.2 Ablation Study

Instructed by the last two motivating questions, we investigate the effect of BeTop formulations and components inside BeTopNet. For efficient study, we randomly partition $20\%$ of WOMD train set for prediction, and directly report the planning results by *Test14-Random* benchmark, which are both representative for the original datasets as verified by [22, 73].

**Synergy with existing state-of-the-art methods.** We first study the effect adjoining BeTop as synergistic objectives over existing SOTA methods in planning and prediction. Described in Table 6 and Table 7, BeTop augments $+2.1\%$ and $2.0\%$ planning score with learning-based and rule-based planners, respectively. Similar compliance effects are also witnessed in guiding strong motion predictors, bringing $+1.1\%, +2.4\%$ improved mAP with $-1.7\%$ prediction errors of minADE.

**Number of interactive agents for topology-guided local attention.** In determining the number $K$ future interactive agents for BeTopNet in local attention, we validate the prediction mAP under an array of agent numbers. Shown in Fig. 4, we observe a converging effect, with maximum $+3.7\%$ mAP by the growing number of interactive agents. A drop of $-1.8\%$ mAP is captured after the peak performance of $K = 32$. It is due to falsely accepting non-interactive agent values by large $K$.

**Different functionalities in BeTopNet.** We further investigate the effects of different functionalities for BeTopNet in Table 8. Compared to the full model, ablations in ID.1 and ID.2 underscore the imitative contingency learning process for costs ($-2.9\%$ CLS) and contingency branching ($-1\%$ CLS-NR). Sole imitative BeTopNet performs the best OLS (ID.2), while the stabilizing effects found in Sec. 4.1 are verified ($-2.8\%$ CLS-NR) in comparing ID.3-ID.5 for joint interactive patterns.

Table 6: **Results of integrating BeTop by strong planning baselines in nuPlan benchmark.**

| Method | nuPlan | | | |
|---|---|---|---|---|
| | OLS ↑ | CLS-NR ↑ | CLS ↑ | **Avg. ↑** |
| PDM [69] | 0.463 | 0.898 | **0.918** | 0.760 |
| **PDM [69] +BeTop** | **0.488** | **0.916** | 0.902 | **0.770** |
| PlanTF [73] | 0.871 | 0.864 | 0.805 | 0.847 |
| **PlanTF [73] +BeTop** | **0.878** | **0.882** | **0.807** | **0.856** |

Table 7: **Results of integrating BeTop by strong prediction baselines in WOMD benchmark.**

| Method | WOMD | | | |
|---|---|---|---|---|
| | minADE ↓ | minFDE ↓ | MR ↓ | **mAP ↑** |
| MTR [22] | 0.6046 | 1.2251 | 0.1366 | 0.4164 |
| **MTR [22] +BeTop** | **0.5941** | **1.2049** | **0.1328** | **0.4249** |
| EDA [53] | **0.5708** | **1.1730** | **0.1178** | 0.4353 |
| **EDA [53] +BeTop** | 0.5742 | 1.1853 | 0.1181 | **0.4407** |

Figure 4: **Results of different interactive agents number for local attention**. We observe a convergence effect for the selection of $K$.

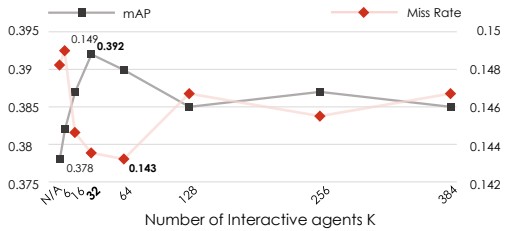

Table 8: **Results of BeTopNet planning performance with different components.** Contingency is the key for closed-loop simulation.

| ID | Ablative Components | nuPlan | | |
|---|---|---|---|---|
| | | OLS ↑ | CLS-NR ↑ | CLS ↑ |
| 0 | BeTopNet | 0.876 | 0.902 | 0.857 |
| 1 | No branched plan | 0.879 | **0.894** | **0.830** |
| 2 | No cost learning | **0.882** | 0.888 | 0.807 |
| 3 | BeTop only | 0.877 | 0.876 | 0.804 |
| 4 | No local attention | 0.871 | 0.852 | 0.804 |
| 5 | Encoders only | 0.867 | 0.827 | 0.784 |

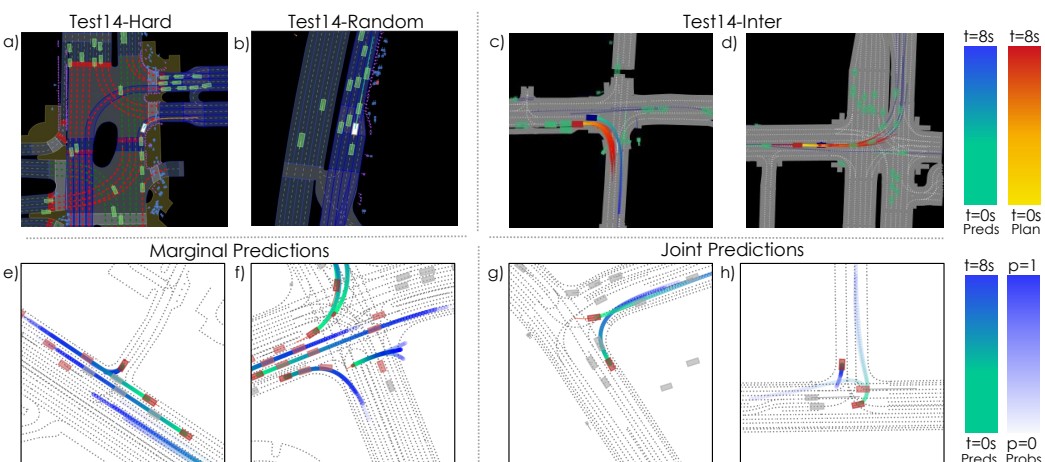

Figure 5: **Qualitative results of planning and prediction in nuPlan and WOMD.** BeTopNet performs compliant reaction simulations in a) yielding for pedestrians; b) cruising in dense traffic. Interactive scenarios (c,d) further present the consistency of contingency learning. BeTopNet predicts both compliant marginal (e,f) and joint (g,h) multi-agent predictions under diverse scenarios. Future interactive behavior patterns can also be consistently reasoned (rendered in light red) with BeTop.

## 5 Conclusion

In this paper, we present BeTop, a topological new-look for multi-agent behavioral formulation. Derived by braid theory, the reasoning tasks for BeTop are drafted supervising joint interactive patterns with integrated prediction and planning. A synergistic network, BeTopNet, is established with an imitative contingency learning process to boost compliant BeTop reasoning. Experiments on nuPlan and WOMD verify BeTopNet's state-of-the-art performance in prediction and planning.

**Limitation and Future work.** Current BeTop considers one-step future topology alone, and focuses on prediction and planning. Future work would be centered on developing a recursive version of BeTop in multi-step, multi-agent reasoning and coordination. Another promising direction would be the connectivity of BeTop upon perceptions as tracking for the end-to-end paradigm, as well as an extension on reasoning behaviors under 3D scenarios for multiple autonomous agents.

## Acknowledgments

This work was supported in part by the Agency for Science, Technology and Research (A*STAR), Singapore, under the MTC Individual Research Grant (M22K2c0079), the ANR-NRF Joint Grant (No.NRF2021-NRF-ANR003 HM Science), the Ministry of Education (MOE), Singapore, under the Tier 2 Grant (MOE-T2EP50222-0002), National Key R&D Program of China (2022ZD0160104), NSFC (62206172), and Shanghai Committee of Science and Technology (23YF1462000).

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

## *Appendix*

# A    Discussions

Towards a better understanding of this work, we supplement intuitive questions that may raise. Note that the following list does *not* indicate the manuscript was submitted to a previous venue or not.

**Q1:** *How does BeTop bridge and discern with dense, sparse, and topological representations?*

BeTop is derived from braid theory [46], reasoning the topology that explicitly labels consensual interactions as dense, occupancy-like intertwines from sparse, braided future trajectories of multi-agent behaviors. Unlike fixed occupancy [14, 21, 18], BeTop dynamically forecasts behavioral interactions by agents collectively, serving as a guiding medium for node reasoning in joint planning and prediction compared to standard sparse predictions [25, 24, 66]. Differentiating from typical topological approaches, BeTop explicitly formulates coordinated multi-agent behaviors and reasons topology in occupancy manners, rather than relying on implicit relation graph learning [41, 67, 40] or complex braid inference [44, 45, 61].

**Q2:** *Why is BeTop allied with contingency instead of conditional or game-theoretic reasoning?*

Contingency plays the most suitable role in BeTopNet, addressing multi-agent uncertainty by deferring uncertain planning with long-term joint predictions from interactive agents under behavioral compliance. This aligns with BeTop's reasoning targets, which aim to achieve a one-shot consensus among all behaviors by formulating future behavior as coordinated joint interactions. This synergy alleviates the challenges led by game-theoretic reasoning [8, 33] or conditional integration [31, 28, 32], which are struggled by multi-step interactive rollouts and unstable joint behavioral patterns.

**Q3:** *What would be the broader impact and future direction?*

BeTop steps the first trial towards an explicit topological formulation and reasoning paradigm for multi-agent interactive behaviors. This serves as a basis in exploring immense interactive behaviors in the real-world, and is reasoned by the autonomous agents jointly in a coordinated manner. For instance, BeTop with enlarged scalability and dimension may summarize the topology for all behaviors in 3D scenarios or even larger spaciality, and may reasoned through end-to-end BeTopNet as a foundation behavioral model. Moreover, we can consider BeTop's capability as collective maneuvers, which can be further leveraged in coordinating naturalistic and efficient decision-making for multiple autonomous agents.

# B    Properties of BeTop

In this section, we supplement additional properties that characterize the formulation of Behavioral Topology (BeTop). Analytically, BeTop is highlighted with: 1) geometric invariant (Theorem B.1); 2) approximated topological invariant (Theorem B.2); and 3) asymmetric topology (Theorem B.3).

**Theorem B.1** (**Geometric Invariant**). The topology results of $\mathcal{E} \subset \mathcal{G}$ in BeTop remain unchanged given arbitrary geometrical transformations for the collective scene trajectories $\mathbf{Y}_n$.

*Proof.* Given arbitrary rotation $\mathbf{H} \in \mathbb{R}^{2 \times 2}$ and shifting $\mathbf{b} \in \mathbb{R}^2$. Consider the mapping $g$ for elementary topology $e_{ij} \in \mathcal{E}$ from future trajectories $(\mathbf{Y}_i, \mathbf{Y}_j)$, $i \neq j \in [1, N_a]$, the local transformation $f : g \to h \circ f$ to $i$'s coordinate is invariant, such that $h(f(\mathbf{Y}_i, \mathbf{Y}_j)) = h(f(\mathbf{H}\mathbf{Y}_i + \mathbf{b}, \mathbf{H}\mathbf{Y}_j + \mathbf{b}))$. Hence, given the function sets $f_j^i \in \sigma_i \subset f, \mathbf{I} \in h$ defined in Sec. 3.1; $e_{ij} \in \mathcal{E}$ is also invariant. $\qquad\square$

*Remark* 1. The Theorem B.1 proves the behavioral stability of BeTop given arbitrary multi-agent trajectories patterns for planning and predictions. Any rotations and movement of the original scene will not interfere with the formulated results of BeTop.

**Definition B.1** (**Topological Invariant**). Given future trajectory pairs $(\mathbf{Y}_i, \mathbf{Y}_j)$, $i \neq j \in [1, N_A]$ with certain current heading $(\theta_i^0, \theta_j^0)$, the sum of future relative angles (winding number) $w_{ij} = \frac{1}{2\pi} \sum_0^{T_f} \Delta\theta_{ij}^t$ form its first-order [87] topological invariant.

*Proof.* Consider the polar representation for the closed form $\psi_i(t) = ||\psi_i(t)||e^{i\theta_i(t)}$. Where $\psi_i : [0, T_m] \to \mathbb{C}\backslash\{0\}, i \in [0, n]$, we can define the winding function $\lambda_i(t) = \frac{1}{2\pi i} \int_{\psi_i} dz/z, z =$

$\psi_i(t), t \in [0, T_m]$. This Cauchy formula [2] can be further integrated as:

$$\lambda_i(t) = \frac{1}{2\pi i} \log(\frac{||\psi_i(t)||}{||\psi_i(0)||}) + \frac{1}{2\pi}(\theta_i(t) - \theta_i(0)). \tag{7}$$

We are interested in the real (first-order) part of $\lambda_i(t)$ which is an invariant topologically. Hence, the trajectory pairs $(\mathbf{Y}_i, \mathbf{Y}_j), 0 < t \leq T_f < T_m$ can be described as: $\mathbf{Y}_i = \psi_i : (0, T_f]$. The joint invariant across future $w_{ij} = \sum_t^{T_f}(\lambda_i(t) - \lambda_j(t))$ then becomes:

$$w_{ij} = \frac{1}{2\pi} \sum_t^{T_f}(\theta_i^t - \theta_j^t) - \frac{1}{2\pi} \sum_t^{T_f}(\theta_i^0 - \theta_j^0). \tag{8}$$

As the current heading pair $(\theta_i^0, \theta_j^0)$ is certain, the invariant becomes $w_{ij} = \frac{1}{2\pi} \sum_0^{T_f} \Delta\theta_{ij}^t$ which proofs the definition. $\square$

**Corollary B.1.1.** Given any $\Delta\theta_{ij}^t \in [-\frac{\pi}{2}, \frac{\pi}{2}]$, where $0 < i.j < N_a, t \in (0, T_f]$, and the constant $\eta_i, \eta_j \in \mathbb{R}$, the transformed $w_{ij}^T = \sum_0^{T_f}(\eta_j \sin \Delta\theta_{ij}^t - \eta_i \sin \theta_i^t)$ is also topological invariant.

*Proof.* The defined function $\sin(\cdot)$ is a monotone mapping under $[-\frac{\pi}{2}, \frac{\pi}{2}]$. Hence, this firstly enables $\sum_0^{T_f} \eta_i \sin \theta_i^t$ uniquely defines $\mathbf{Y}_i$. More than that, $w_{ij}^T$ is the unique mapping value of $w_{ij}$ with $\mathbf{Y}_i$ under defined transformation, and thereby keep the invariant property. $\square$

**Theorem B.2.** The edge topology $\mathcal{E} \subset \mathcal{G}$ is an approximate of topological invariant, so that $e_{ij} \in \mathcal{E}, 0 < i, j \leq N_a$ is characterized by $w_{ij}$.

*Proof.* Given future trajectories $\mathbf{Y}_i, \mathbf{Y}_j$, we consider the braid functions $\sigma_i$ maps monotonically increased transformations $f_i^i, f_j^i$ to $i$'s local coordinate,as defined in Sec. 3.1. We assume a continuous future horizon $(0, T_f]$ where the headings for $f_j^i(\mathbf{Y}_j)$ is defined by relative angles $\Delta\theta_{ij}(t)$. Thereby, the transformed lateral trajectory for agent $j$ can be formed as: $\int_0^t \eta_j \sin \Delta\theta_{ij}(t)dt$. Similarly, $f_i^i(\mathbf{Y}_i)$ can be formed as $\int_0^t \eta_j \sin \theta_i(t)dt$, where $\eta_i, \eta_j$ denotes small constant step lengths.These form the original intersection function $\mathbf{I}$ in Sec. 3.1 as:

$$\mathbf{I}\left(f_i^i(\mathbf{y}_i^t), f_j^i(\mathbf{y}_j^t)\right) \rightarrow \int_0^t (\eta_j \sin \Delta\theta_{ij}^t - \eta_i \sin \theta_i^t)dt, \tag{9}$$

where $\mathbf{I}(\cdot) = 0$ denotes braid intertwine for interactive behaviors. As the term in Corollary B.1.1 of $w_{ij}^T$ (the sum of the right term) is an approximate (discretization) of the right term, this proves the edge topology $e_{ij} \in \mathcal{E} \rightarrow \max_{T_f} \mathbf{I}(\cdot)$ as the approximation of topological invariant. $\square$

*Remark* 2. The Theorem B.2 proves the generality of BeTop in terms of future interactive behavioral patterns. The approximated topological invariant property prompts a representative of various future states sharing similar behavioral or identical interactive patterns by BeTop.

**Theorem B.3 (Asymmetric Topology).** Edge topology $\mathcal{E} \subset \mathcal{G}$ is not symmetric such that $\exists i, j, e_{ij} \not\equiv e_{ji}, 0 < i, j \leq N_a$.

*Proof.* Given Eq. (9) defined in Theorem B.2,we can always construct a case $\exists t_i, t_j \in (0, T_f]$, the intersection $\mathbf{I}\left(f_i^i(\mathbf{y}_i^{t_i}), f_j^i(\mathbf{y}_j^{t_i})\right) = 0$, but $\mathbf{I}(f_i^j(\mathbf{y}_j^{t_j}), f_i^j(\mathbf{y}_i^{t_j})) \neq 0$, which prove the claim. $\square$

*Remark* 3. The Theorem B.3 proves a more naturalistic interactive behavior of BeTop. It is likely in a real-world scenario that the future behavior of agent $A_i$ is interacted by agent $A_j$, while $A_j$ does not.

**Computational complexity.** The complexity in computing full $\mathcal{E}$ is $\mathcal{O}(N_a^2)$. In practice, we downscale the sourced as the agents of interests $N_I < N_a$, such that $\mathcal{O}(N_I N_a)$. It is much less than braid sequence inference [45] with maximum $\mathcal{O}((N_a - 1)^{N_a})$ computational costs. Further analytical proof of computational efficiency leveraging braids can be found in [44].

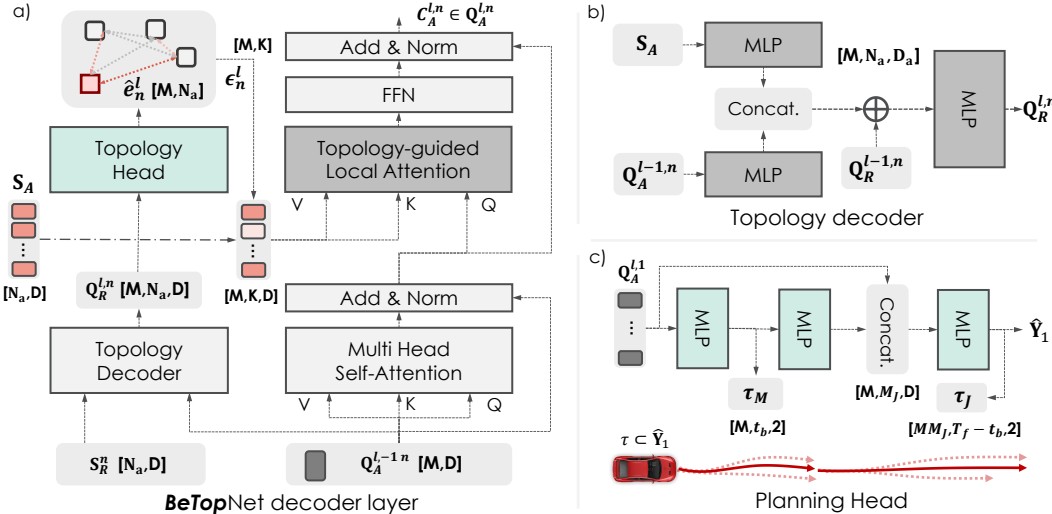

Figure 6: **Structural details in BeTopNet.** a) The learning structure of single synergistic decoder layer featuring `TransDecoder` and `TopoAttn` design; b) The structure inside topology decoder network of `TopoDecoder`; c) Branched planning head design corresponding to contingency planning.

## C Implementation Details

In this section, we instantiate further details for BeTop on the configurations for BeTopNet structure, and provide contingency learning paradigms for both prediction and planning challenges.

### C.1 Model Structure

Subjecting to different testing requirements defined in nuPlan and WOMD, we set up two model variants for BeTopNet in formulating planning and prediction challenges. Apart from topology reasoning for interactive behaviors, for planning challenges, BeTopNet integrates both tasks of prediction and planning. In the prediction challenges under marginal and joint settings, BeTopNet is allocated only with the prediction parts. The structural details are illustrated below.

**Scene inputs.** Carving the driving scenarios involves historical agent states $\mathbf{X}$ and map polylines $\mathbf{M}$ as scene inputs. For planning settings, we collect scene agent states with past $T_h = 2$ seconds at 10Hz, leaving basic kinematic states as $(x, y, v_x, v_y, \theta)$, joining with agent shapes and types. We only keep the current state for ego vehicle (AV) in preventing closed-loop gap [73, 69, 88] with open-loop training as recently discussed in the community. $N_m = 256$ segments of the map with length $L_m = 20$ are gathered by scene-centric manners considering positions, traffic lights, and speed limits. The prediction task is built on WOMD considering $T_h = 1$ seconds at 10Hz with scenarios of larger scalability. It is followed by full scene agents with $N_m = 768$ map segments of identical states for both $\mathbf{X}$ and $\mathbf{M}$.

**Scene encoder.** Both scene attributes $\mathbf{X}, \mathbf{M}$ are firstly encoded leveraging layered point encoder [89] to hidden dimension $D$ shared throughout the BeTopNet structures, with $D = 128$ for planning and $D = 256$ for prediction tasks. A stack of Transformer encoders is then devised with 4 layers in planning and 6 in prediction for $\mathbf{S}_A, \mathbf{S}_M$. Due to scalable settings for prediction tasks, local attention with the nearest 16 keys is built in each layer. Following [22, 73], the dense prediction head is adopted for all agents after the encoder, enhancing future semantics.

**Synergistic decoder.** Depicted in Fig. 6, the decoder structure is founded by an iterative stack of $L$ Transformer decoders querying $M$ modes of future trajectories $\hat{\mathbf{Y}}$ with dual stream in reasoning topology $\hat{\mathcal{E}}$. Consisting of $L = 6$ and $L = 4$ decoders for prediction and planning respectively, it is initialized jointly by relative features $\mathbf{Q}_R^{0,n}$ and decoding queries $\mathbf{Q}_A^{0,n}$. Relative attributes $\mathbf{S}_R$ are computed efficiently following [64] for relative distance and headings. $M = 6$ learnable embedding are devised as $\mathbf{Q}_A^{0,n}$ for planning, and we utilize the anchored ones [22] with $M = 64$ in prediction tasks. As displayed in Fig. 6(a), dual queries $\mathbf{Q}_A^{j-1,n}, \mathbf{Q}_R^{j-1,n}$ are served from the last level. The

decoding process (following Eq. (3)) iteratively reasons $\hat{e}_n^l$ from `TopoDecoder`, serving as a prior guiding agent semantics by `TopoAttn` inside `TransDecoder`, which concurrently aggregates scene semantics from agents $\mathbf{S}_A$ and maps $\mathbf{S}_M$. Expressly, the structure of `TopoDecoder` (Fig. 6(b)) comprises simple MLPs and update $\mathbf{Q}_R^{l,n}$ by concatenation sourcing query features $\mathbf{Q}_A^{j-1,n}$ with agent semantics $\mathbf{S}_A$, and connect residuals $\mathbf{Q}_R^{l-1,n}$ from last layer. Decoding queries $\mathbf{Q}_A^{l,n}$ is updated by a concatenation from aggregated agent feature $\mathbf{C}_A^{l,n}$, map features $\mathbf{C}_M^{l,n}$, and agent semantics $\mathbf{S}_A^n$ directly from encoder. Agent feature $\mathbf{C}_A^{l,n}$ is aggregated by `TopoAttn`, where the local attention is devised using deployments from [90] indexing $K = 32$ agents from reasoned topology $\hat{e}_n^l$. We omit the aggregation process for map features, which performs the vanilla Transformer decoder structure for planning and dynamic collection form by [22] under prediction tasks for hefty map features.

**Reason heads.** Following the contents in Sec. 3.2, reason heads in decoding prediction and topology follow simple MLPs given respective decoding queries. For the planning head, it leverages a cascaded design for branched contingency planning with multi-modalities (Fig. 6(c)). Specifically, the short-planning $\tau_M$ is decoded by the AV future states $\{\hat{y}_1^{1:t_b}\}_m$ from first stage head with $m \in M$ modes, where $t_b = 3$ denotes the branching time. They are then detached and leveraged as prior for the branched planning. Successive MLPs project and reshape the short-term contingency prior by $\mathbb{R}^{M \times M_J \times D}$ for $M_J = 6$ branches planning $\mathcal{T}_J^m$ under each of $\tau_M^m$. Further concatenated by broadcasted decoding queries, the planning head generates $M \cdot M_J$ trajectories $\hat{\mathbf{Y}}_1$ for AV.

## C.2 Imitative Contingency Learning

**Efficient joint prediction recombination.** Retrieving the top-performed joint predictions from full marginal predictions $\hat{Y}_M$ sequentially is time-consuming with exponentially complexity. Hence, we firstly downscale the potentially interested agents $N_I$ by sorting the AV-reasoned topology $\hat{e}_1^L$ with the largest $K_M = 4$ value as index given the planning task. For the joint prediction task, $N_I = 2$ is annotated already from the original data. Then, we leverage the tensor broadcasting mechanism in efficiently retrieving $M = 6$ largest joint distributions $P(\hat{Y}_J^M)$ and joint trajectories $\hat{Y}_J^M$ from $N_I$ interacted agents. Given a tensor $\mathbf{P}_J \in \mathbb{R}^{M^{N_I}}$ initialized by ones, the joint score is computed by $N_I$ times of iterative broadcasting $\hat{\mathbf{p}}_n$ on the $n$-th dimension for $\mathbf{P}_J$ as $P_J = \max_M \prod_n^{N_I} \mathbf{P}_J^{(n)} \otimes \hat{\mathbf{p}}_n$. This process only costs 1.6ms in computing $N_I = 4$ joint predictions for contingency planning.

**Imitative contingency objectives.** Followed by learning objectives derived in Sec. 3.3, the imitation objectives for each layer can be represented as $\mathcal{L}_{\text{IL}} = \mathcal{L}_{\mathcal{V}} + \lambda_1 \mathcal{L}_{\mathcal{E}}$. $\lambda_1 = 50$ weighting BCE loss for edge topology reasoning, the NLL loss for $\mathcal{L}_{\mathcal{V}}$ is formulated as:

$$\mathcal{L}_{\text{NLL}} = \log \sigma_x + \log \sigma_y + \frac{\log(1-\rho^2)}{2} + \frac{1}{2(1-\rho^2)} \left( \left(\frac{d_x}{\sigma_x}\right)^2 + \left(\frac{d_y}{\sigma_y}\right)^2 - 2\rho \frac{d_x d_y}{\sigma_x \sigma_y} \right) - \log p(m^*), \quad (10)$$

where $d_x, d_y$ denotes the difference with ground-truths. In determining the component $m^*$, we leverage a winner-take-all (WTA) strategy [91] in planning by measuring the average displacements (ADE) with groung-truths. For prediction tasks, $m^*$ is selected from the closest anchor as in [53]. For the learnable cost functions $\max C_M(\cdot), C_J(\cdot)$ in contingency planning, we leverage the repulsive potential field [8] delineating planning with prediction by $\phi = \min_d 1/(1 + d(\tau, \hat{\mathbf{y}}))$. For $\max C_M(\cdot)$, $\phi$ is gathered across $T_f$ considering the worst case under full marginal prediction $\hat{Y}_M$ comprising $N_a = 32$ scene agents. For the branched cost $C_J(\cdot)$, $\phi$ for each branch is computed considering joint prediction from $N_I = 4$ agents. Following the objective defined in Eq. (6), the learnable contingency cost is defined as: $\mathcal{L}_{\text{CL}} = C_M + \sum_m^M P(\hat{Y}_J^m)C_J^m$. Hence, the general objectives for planning become:

$$\mathcal{L} = \mathcal{L}_{\mathcal{V}} + \lambda_1 \mathcal{L}_{\mathcal{E}} + \lambda_2 \mathcal{L}_{\text{CL}}, \quad (11)$$

where $\lambda_2 = 5$ is the contingency costs weight. Prediction tasks are updated only by $\mathcal{L}_{\text{IL}}$.

**Inference.** Different from the training process, for the planning task we directly select the full planning trajectory of $T_f = 8$ seconds by highest scoring $\tau^* = \text{argmax}_C \hat{\mathbf{Y}}_1$, subjecting to the original task settings in nuPlan. The scoring results are a combination from original confidence $\hat{\mathbf{p}}_1$ and the short-term cost $C_M$ [69]: $C = \hat{\mathbf{p}}_1 + \lambda_m C_M$, where $\lambda_m = 0.5$ facilitates short-term planning compliance [92]. For the prediction task, a post-processing module following [53] is leveraged in selecting $M = 6$ marginal or joint trajectories of $T_f = 8$ seconds among 3 agent types in WOMD.

# D  Experimental Setup Details

In this section, we provide extra details demonstrated in Sec. 4 for the experiment setups, including detailed settings for the proposed benchmark, testing metrics, state-of-the-art baselines, and training.

## D.1  Planning on nuPlan

**Testing metrics.** For open-loop planning tests, the open-loop score (OLS) serves as the general statistics weighting displacement metrics and miss rates. For closed-loop simulations, both metrics (CLS, CLS-NR) are weighted by a series of statistics measuring 1) driving safety, 2) planning progress, 3) driving comforts, and 4) rule obeying. The PDMScore [93, 94] compared in Table 3 is basically a replica of the closed-loop score for efficient computations. It is denoted as:

$$\text{PDM}_{\text{Score}} = \text{CA} \cdot \text{DAC} \cdot \text{DDC} \cdot \frac{w_1\text{TTC} + w_2\text{DC} + w_3\text{EP}}{\sum w_i}, \tag{12}$$

where the sub-metrics are abbreviations referred in Table 3. All general metrics range from 0 to 1.

**Test14-Inter.** We launch the *Test14-Inter* benchmark in verifying the planning systems under typical corner cases containing rich social interactions, or dynamic profiles by complex map forms. This is highly motivated by the issues raised in [69, 88], that massive scenarios may also be completed by a simple motion model. Specifically, we adopt a mining heuristic defining corner cases by which human experts excel but the motion model (constant acceleration vehicle, CAV) fails. For efficient mining, we directly assess planning results by PDMScore and define the criteria as:

$$(\text{PDM}_{\text{Score}}\text{CAV} < \gamma) \wedge (\text{PDM}_{\text{Score}}\text{Expert} \geq (1 - \gamma)), \tag{13}$$

where $\gamma = 0.1$ denotes a scoring threshold for cases that cannot be easily solved by regular motion profiles of the planning maneuvers. As future work, we aim to explore more interactive scenarios aggravating by BeTop as an enhancement.

**Val14.** In pursuing comprehensive comparisons with current methods, we also manifest BeTopNet in the *Val14* set proposed in [69]. It is a subset of 1040 scenes from the validation set. However, since a portion of validation scenes are shared with the training set in nuPlan [80], we argue this is less representative of testing fairness for learning-based methods. Hence, we only place it as supplementary.

**Baselines.** For all baselines presented in the planning task, we directly report their previous benchmark results. Additional results in the proposed benchmark (Table 3) and ablation studies (Table 6) are re-implemented by the official releases [69, 72, 47, 69]. Expressly, we study the state-of-the-art planning systems categorized by: 1) Rule-based: performing maneuvers by designate rules with the reactive agents [70] or mimicking the planning score [69]; 2) Hybrid: incorporating rules [69] or post-optimizations [8] with a learning-based model; and 3) Learning-based: end-to-end planning with GNN [83, 74] or Transformer [72, 73] enabled models, as well as concurrent methods [92] augmented by representation learning. For ablation studies in Table 6, BeTop is trained directly by the proposed topology decoder with the original PlanT [73] pipeline. For the PDM [69] as a rule-based planning system, we integrate BeTop by replacing the original rule-based motion model with predictions generated from BeTopNet.

## D.2  Prediction on WOMD

**Testing metrics.** For the prediction task, the mean AP (mAP) and Soft-mAP scores are assigned as the primary metrics in computing multi-modal predictions modeled by marginal or joint distributions [35, 81, 82]. Displacement metrics of minADE and minFDE provide the multi-modal prediction errors closest to ground truths without considering the prediction scores.

**Baselines.** We also directly provide the prediction results displayed on the official leaderboards in Tables 4 and 5. Ablation studies in Table 7 are reproduced by the official codes [22, 53]. The prediction performance of BeTopNet is compared against SOTA baselines by: 1) GNN-enabled interactive graph [37, 24]; 2) conditional or game-theoretic behavioral interactions [31, 8]; 3) DETR-based Transformer attentions [22, 25, 53, 86]; and 4) auto-regressive modeling [32].

Table 9: **Detailed nuPlan closed-loop simulation results in Val14 benchmark.** BeTopNet highlights leading results among SOTA methods in safety and compliance, outperforms learning-based agents.

| Type | Method | | | | | Val14 | | | |
| --- | --- | --- | --- | --- | --- | --- | --- | --- | --- |
| | | CA ↑ | TTC ↑ | DDC ↑ | DC ↑ | EP ↑ | Speed ↑ | **CLS-NR ↑** | **CLS-R ↑** |
| *Expert* | *Log Replay* | 0.987 | 0.944 | 0.981 | 0.993 | 0.989 | 0.965 | 0.937 | 0.812 |
| Rule | IDM [70] | 0.909 | 0.834 | 0.941 | 0.944 | 0.862 | 0.973 | 0.793 | 0.793 |
| | PDM-Closed [69] | 0.981 | 0.933 | 0.998 | 0.955 | 0.921 | 0.998 | 0.932 | 0.930 |
| Hybrid | GameFormer [8] | 0.943 | 0.867 | 0.948 | 0.933 | 0.890 | 0.987 | 0.829 | 0.838 |
| Learning | UrbanDriver [74] | 0.856 | 0.803 | 0.908 | **1.000** | 0.808 | 0.915 | 0.677 | 0.648 |
| | PDM-Open [69] | 0.745 | 0.691 | 0.879 | 0.995 | 0.698 | 0.977 | 0.502 | 0.548 |
| | PlanCNN [72] | 0.869 | 0.814 | 0.850 | 0.814 | 0.806 | 0.980 | 0.669 | 0.646 |
| | GC-PGP [83] | 0.858 | 0.801 | 0.897 | 0.900 | 0.603 | **0.993** | 0.611 | 0.549 |
| | PlanTF [73] | 0.941 | 0.907 | 0.968 | 0.937 | **0.898** | 0.977 | 0.853 | 0.771 |
| | PLUTO [92] | 0.961 | **0.933** | 0.985 | 0.964 | 0.895 | 0.981 | **0.890** | 0.800 |
| | **BeTopNet (Ours)** | **0.966** | 0.916 | **0.995** | 0.932 | 0.866 | 0.971 | 0.883 | **0.837** |

Table 10: **Detailed nuPlan closed-loop planning results (PDMScore) on Test14-Random benchmark.**

| Type | Method | | | Test14 Random | | | | |
| --- | --- | --- | --- | --- | --- | --- | --- | --- |
| | | Col. Avoid ↑ | Drivable ↑ | Direction ↑ | Progress ↑ | TTC ↑ | Comfort ↑ | **PDMScore ↑** |
| *Expert* | *Log Replay* | 0.996 | 0.962 | 0.996 | 0.664 | 0.985 | 1.000 | 0.832 |
| Rule | PDM-Closed [69] | 0.934 | 0.984 | 0.996 | 0.867 | 0.911 | 0.996 | 0.888 |
| Learning | Constant Acc. | 0.846 | 0.907 | 0.915 | 0.436 | 0.804 | 1.000 | 0.592 |
| | UrbanDriver [74] | 0.965 | 0.961 | 0.986 | 0.611 | 0.957 | 1.000 | 0.788 |
| | PlanCNN [72] | 0.935 | 0.938 | 0.971 | 0.591 | 0.888 | 0.989 | 0.736 |
| | PlanTF [73] | 0.966 | 0.948 | 0.625 | 0.626 | 0.918 | 0.992 | 0.768 |
| | **BeTopNet (Ours)** | **0.989** | **0.977** | **0.989** | **0.673** | **0.969** | **1.000** | **0.833** |

Table 11: **Detailed nuPlan closed-loop planning results (PDMScore) on Test14-Hard benchmark.**

| Type | Method | | | Test14 Hard | | | | |
| --- | --- | --- | --- | --- | --- | --- | --- | --- |
| | | Col. Avoid ↑ | Drivable ↑ | Direction ↑ | Progress ↑ | TTC ↑ | Comfort ↑ | **PDMScore ↑** |
| *Expert* | *Log Replay* | 0.985 | 0.945 | 0.970 | 0.658 | 0.955 | 1.000 | 0.786 |
| Rule | PDM-Closed [69] | 0.933 | 0.952 | 0.976 | 0.779 | 0.852 | 0.981 | 0.811 |
| Learning | Constant Acc. | 0.845 | 0.871 | 0.861 | 0.415 | 0.800 | 1.000 | 0.552 |
| | UrbanDriver [74] | 0.946 | 0.944 | 0.992 | 0.581 | 0.903 | **1.000** | 0.731 |
| | PlanCNN [72] | 0.909 | 0.908 | 0.937 | 0.555 | 0.860 | 0.992 | 0.675 |
| | PlanTF [73] | **0.984** | **0.961** | **0.996** | 0.649 | **0.961** | 0.996 | 0.813 |
| | **BeTopNet (Ours)** | 0.968 | 0.945 | 0.972 | **0.747** | 0.908 | 0.996 | **0.813** |

## D.3 Training Setup

BeTopNet for both prediction and planning tasks are trained in end-to-end manners by AdamW optimizer with 4 NVIDIA A100 GPUs. The learning rate is configured as $1e^{-4}$ scheduled with the multi-step reduction strategy. The planning model is trained by 25 epochs with a batch size of 128, while the prediction task is trained with 30 epochs with a batch of 256.

# E Additional Quantitative Results

## E.1 Planning

**Additional planning results in Val14.** We evaluate the closed-loop simulation performance under *Val14* in Table 9, BeTopNet hovers strong planning results and is comparable ($+4.6\%$ CLS) to concurrent learning-based methods [92] leveraging extra contrasting learning for training augmentations. BeTopNet is also featured by leading driving safety ($+2.7\%$ CA, $+1.0\%$ TTC) and compliance ($+2.8\%$ DDC) compared with other strong models [73, 8]. However, due to the data leakage of *Val14* with training set by a part of shared scenarios, we only provide the results as a reference.

**Additional planning effects in Test14.** To delve into the planning results of BeTopNet, we present statistics measuring by another detailed metric, PDMScore, for both of the *Test14* benchmarks in Table 2. Exhibited in Tables 10 and 11, BeTopNet delivers strong maneuver safety and compliance, marking solid PDMScore from both benchmarks. Compared with learning-based methods, BeTopNet

Table 12: **Marginal predictions on WOMD Motion Leaderboard [81].** Primary metric.

| Category | Method | minADE ↓ | minFDE ↓ | Miss Rate ↓ | mAP ↑ | **Soft mAP ↑** |
|---|---|---|---|---|---|---|
| Vehicle | MTR [22] | 0.7642 | 1.5257 | 0.1514 | 0.4494 | 0.4590 |
| | EDA [53] | **0.6808** | 1.3921 | **0.1164** | 0.4833 | 0.4972 |
| | **BeTopNet (Ours)** | 0.6814 | **1.3888** | 0.1172 | **0.4860** | **0.4995** |
| Pedestrian | MTR [22] | 0.3486 | 0.7270 | 0.0753 | 0.4331 | 0.4409 |
| | EDA [53] | **0.3426** | **0.7080** | 0.0670 | 0.4680 | 0.4778 |
| | **BeTopNet (Ours)** | 0.3451 | 0.7142 | **0.0668** | **0.4777** | **0.4875** |
| Cyclist | MTR [22] | 0.7022 | 1.4093 | 0.1786 | 0.3561 | 0.3650 |
| | EDA [53] | 0.6920 | 1.4106 | **0.1673** | 0.3947 | 0.4037 |
| | **BeTopNet (Ours)** | **0.6905** | **1.3975** | 0.1688 | **0.4060** | **0.4163** |

Table 13: **Joint predictions on WOMD Interaction Leaderboard [82].** Primary metric.

| Category | Method | minADE ↓ | minFDE ↓ | Miss Rate ↓ | **mAP ↑** | Soft mAP ↑ |
|---|---|---|---|---|---|---|
| Vehicle | GameFormer [8] | 1.0499 | 2.4044 | 0.4321 | 0.2469 | 0.2564 |
| | AMP [32] | **0.9862** | **2.2286** | **0.3726** | 0.3104 | 0.3196 |
| | **BeTopNet (Ours)** | 1.0216 | 2.3970 | 0.3738 | **0.3374** | 0.3308 |
| Pedestrian | GameFormer [8] | 0.7978 | 1.8195 | 0.4713 | 0.1962 | 0.2014 |
| | AMP [32] | **0.6823** | **1.5244** | **0.3716** | **0.2359** | **0.2423** |
| | **BeTopNet (Ours)** | 0.7862 | 1.8412 | 0.4074 | 0.2212 | 0.2267 |
| Cyclist | GameFormer [8] | 1.0686 | 2.4199 | 0.5765 | 0.1367 | 0.1338 |
| | AMP [32] | **1.0533** | **2.3715** | **0.5194** | 0.1420 | 0.1477 |
| | **BeTopNet (Ours)** | 1.1155 | 2.5850 | 0.5253 | **0.1717** | **0.1756** |

excels in closed-loop driving progress (+15.1%, +7.5% EP), safety (+5.6% TTC, +2.9% CA), and the general score (+8.5% PDMScore). For rule-based systems, the leading performance is empirically by virtue of a constant driving progress. This may refer to an unresolved *copy-cat* problem [73] for imitative planners. It requires further integration and fallback with rule-based methods for on-board AD system design in practice.

### E.2 Prediction

**Per-category marginal prediction.** In Table 12, We mainfest the prediction performance of BeTop-Net under each prediction category. Compared against the concurrent SOTA motion predictors [53], BeTopNet demonstrates superior mAP-based metrics among all types for compliant predictions. Specifically, overall improvements in Cyclist denote refined interactive patterns captured by BeTop, as the cyclist predictions are the most uncertain task with less reliance on map information.

**Per-category joint prediction.** We further instantiate the per-category joint prediction of BeTopNet with SOTA methods in Table 13. Compared with concurrent methods [32] featuring auto-regressive decoding, BeTopNet achieves robust displacement metrics, while outperforming in prediction compliance of mAP metrics (+8.7%, +20.9% mAP) due to advanced joint modality scoring stabilized by edge topology in BeTopNet. Moreover, the coordinated joint behaviors reasoned by BeTopNet largely mitigate the unstable patterns against game-theoretic method [8] (−15.6%, −15.7%, −9.7% Miss Rate) under similar model architecture.

## F Additional Ablation Studies

**Scaling effects of model and decoding agents.** The scalability challenges begin with the scaling of our BeTopNet models to accommodate varying scene agents and map. Experimentally, we configure BeTopNet with different model scales to evaluate whether our approach maintains its effectiveness.

In Table 14, BeTopNet is evaluated by three model scales varying in decoding modalities and dimensions. The results demonstrate that BeTopNet consistently improves prediction accuracy, with an increase from 0.391 to 0.442 (+13.4% mAP) and a decrease in the Miss Rate (−11.9%). This showcases its enhanced robustness in handling multi-agent settings by enlarging model scales.

In Table 15, BeTopNet reports comparable computational costs compared to [22], while with better prediction accuracy shown in Table 12. The similar latency is due to the topo-guided attention,

Table 14: **Effects of varied model scale.** BeTopNet shows scalability with the number of decoding modalities and feature dimensions.

| Scale | mAP ↑ | Miss Rate ↓ | Latency (ms) | # Params. (M) |
|---|---|---|---|---|
| Small | 0.391 | 0.131 | 45 | 28.91 |
| Medium | 0.437 | 0.119 | 65 | 28.91 |
| Base | **0.442** | **0.117** | 70 | 45.38 |

Table 15: **Effects of varied decoding agents.** Computational costs of [22] are reported in the parenthesis after ours.

| # Decoding Agents | Latency (ms) | GPU Memory (G) |
|---|---|---|
| 8 | 89 (84) | 6.5 (5.2) |
| 16 | 120 (123) | 10.8 (7.1) |
| 32 | 166 (193) | 19.2 (15.6) |

Table 16: **Effects of varied temporal granularity in BeTop.** Future interactions are split into various intervals for multi-step BeTop labels. A fine-grained topology reasoning for the whole prediction horizon results in a slightly improved performance and increased computational costs simultaneously.

| Interval | minADE ↓ | minFDE ↓ | Miss Rate ↓ | mAP ↑ | Inference Latency (ms) | Training Latency (ms) | # Params. (M) |
|---|---|---|---|---|---|---|---|
| 1(Base) | 0.637 | 1.328 | 0.144 | 0.392 | 70.0 | 101.6 | 45.380 |
| **2** | **0.633** | **1.325** | 0.145 | **0.394** | 75.5 | 110.6 | 45.382 |
| 4 | 0.634 | 1.326 | **0.142** | 0.391 | 80.0 | 133.4 | 45.386 |
| 8 | 0.641 | 1.347 | 0.147 | 0.389 | 90.0 | 255.0 | 45.393 |

Table 17: **Effects of varied model foundation by Wayformer [62].** Synergistic decoder design by BeTopNet demonstrate solid multi-agent interaction understanding compared with vanilla design.

| Method | minADE ↓ | minFDE ↓ | Miss Rate ↓ | mAP ↑ |
|---|---|---|---|---|
| Wayformer | 0.661 | 1.417 | 0.199 | 0.281 |
| Wayformer+BeTop | 0.637 | 1.364 | 0.178 | 0.290 |
| Wayformer+BeTopNet | **0.604** | **1.261** | **0.166** | **0.344** |

which reduces the KV features in agent aggregation during decoding. While BeTop introduces extra computations for reasoning, it requires more GPU memory for cached topology tensors.

**Temporal granularity in BeTop.** Table 16 explores the effect of varied temporal granularity in BeTop, with minimal adjustments to BeTopNet. In our study, future interactions are split into multi-step BeTop labels. Topology reasoning task is then deployed through expanded MLP Topo Head for output steps. Compared to the baseline 1-step reasoning, multi-step BeTop reasoning slightly improves performance (*e.g.*, 2-steps, +0.2 mAP), with a corresponding increase in computational costs for additional steps. This highlights the potential of multi-step reasoning to enhance BeTopNet in interactive scenarios, while refining temporal granularity for more accurate and efficient interactions remains an open question. We believe how to effectively leverage multi-step BeTopNet represents an interesting area for future exploration.

**Synergy with additional model foundation.** To understand the generalization under different model foundations, we conduct additional ablations integrating BeTop with reproduced Wayformer [62] in [95]. As reported in Table 17, incorporating BeTop as supervision improves vanilla Wayformer with a $-6.2\%$ Miss Rate and $+3.2\%$ mAP. Furthermore, integrating BeTopNet significantly boosts performance, achieving a $+18.6\%$ mAP and $-7.2\%$ Miss Rate. This enhancement is due to synergistic decoder design, which uses iterative BeTop reasoning and Topo-guided attention to refine trajectories by selectively aggregating interactive features.

## G Additional Qualitative Results

**Additional planning results.** We provide the qualitative closed-loop simulations for all of the benchmarks in *Test14*, as shown in Figs. 7 and 8.

**Additional prediction results.** We provide the qualitative prediction results for BeTop with reasoned edge topology under both marginal (Fig. 10) and joint (Fig. 9) challenge settings.

## H License of Assets

Data for nuPlan [80] and WOMD [35] are complied with CC-BY-NC 4.0 licence and Apache License 2.0; The code for re-implementations are under Apache License 2.0 for PDM [69] and MTR [22], and

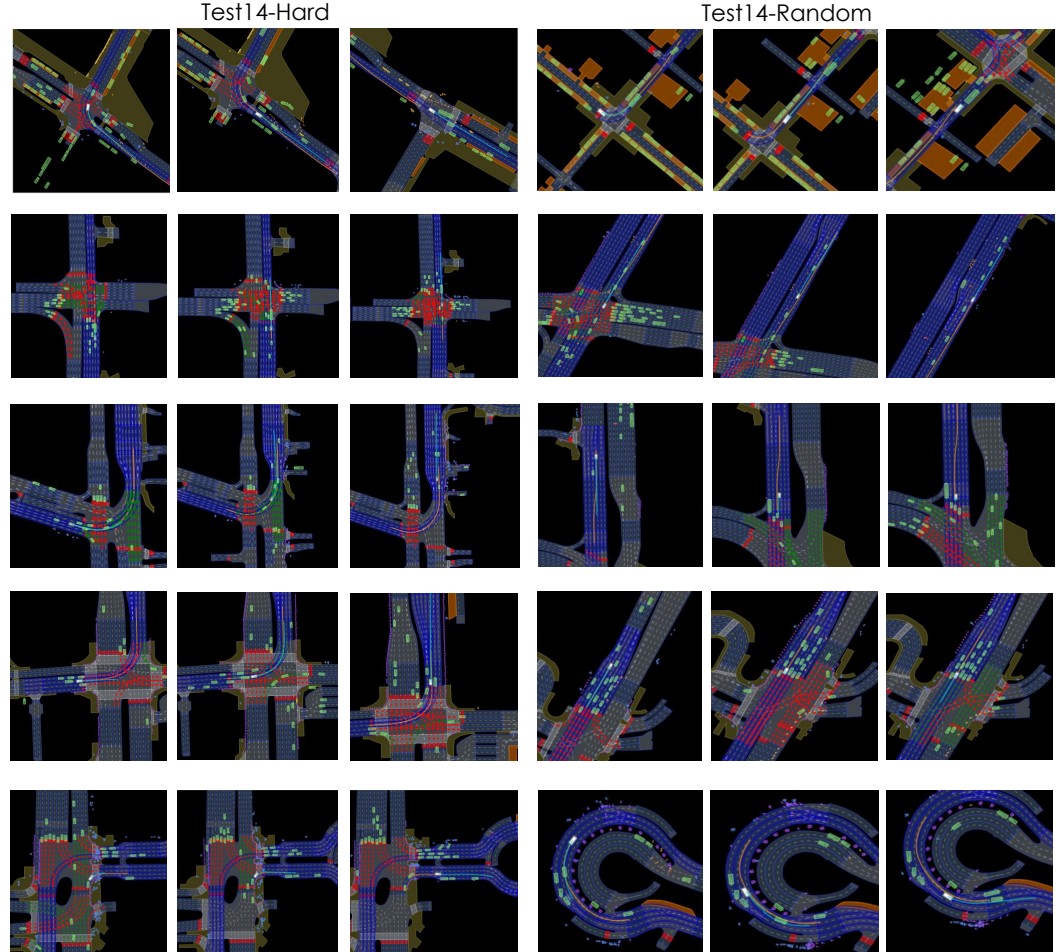

Figure 7: **Qualitative results of BeTopNet in nuPlan planning under Test14 simulations.** Each row of the figures render closed-loop simulations at 1s, 8s, and 15s temporal frames. As illustrated, BeTopNet performs consistent planning under challenging driving scenarios of diverse categories.

MIT License for EDA [53] and PlanTF [73], respectively. The source code and our trained models will be publicly available under the Apache License 2.0.

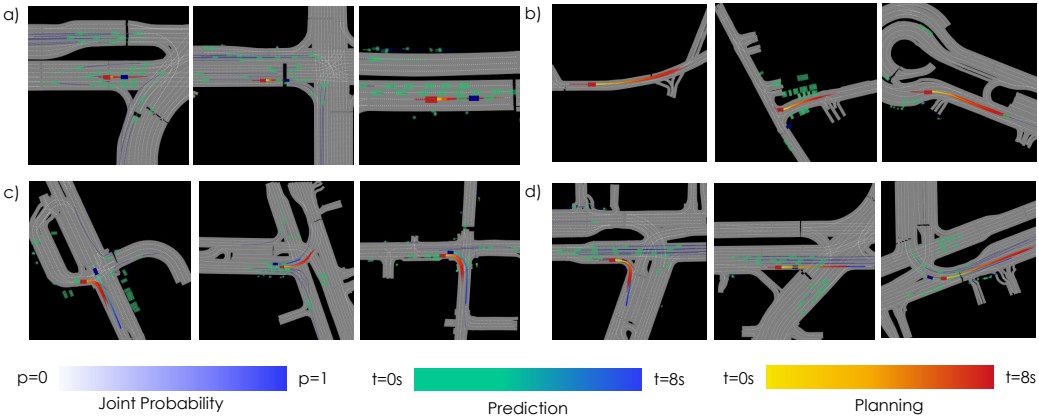

Figure 8: **Qualitative results of BeTopNet in nuPlan planning under Test14-Inter.** BeTopNet performs compliant planning under: (a) yielding to front agents; (b) cruising on various road structure; (c-d) interactive behaviors among two or more agents with dense traffic.

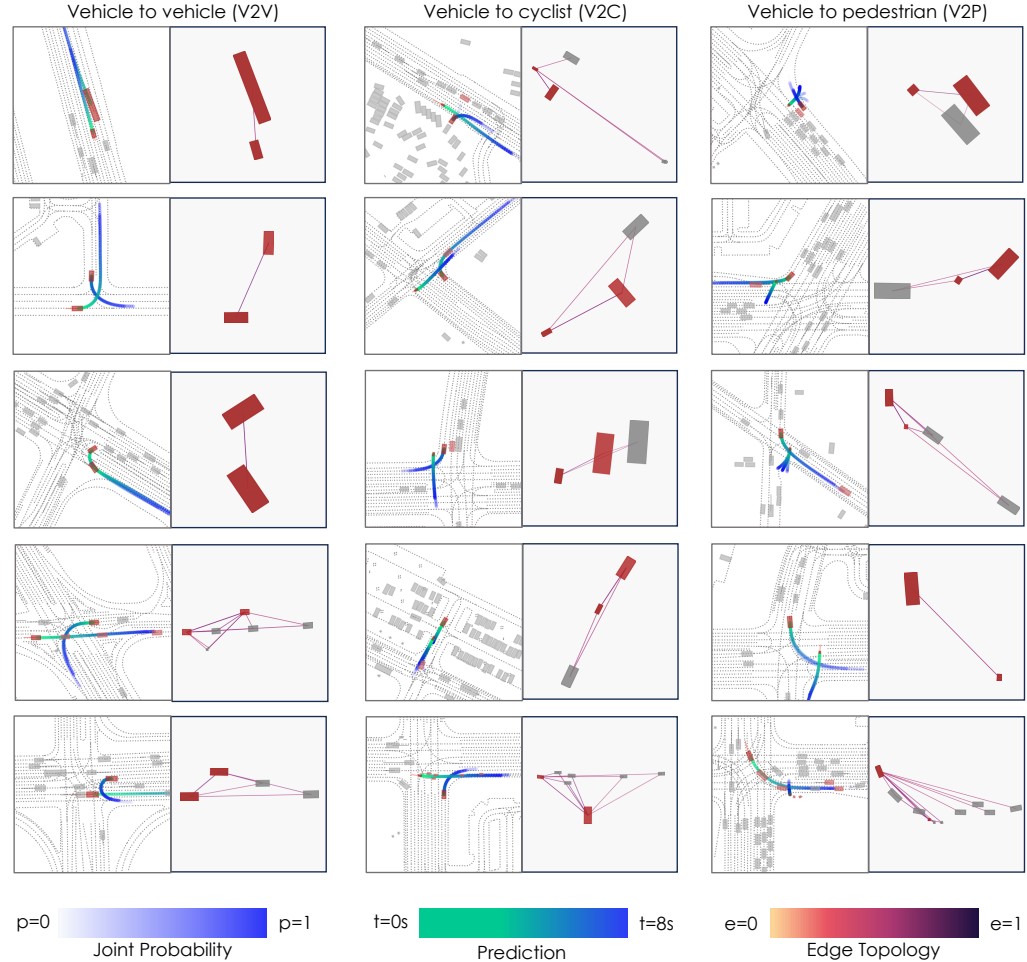

Figure 9: **Qualitative results of BeTopNet in WOMD joint prediction.** Joint predictions among heterogeneous agents are categorized by each column (V2V, V2C, and V2P) with corresponding TopK reasoned topology. As depicted, BeTopNet can accurately capture the future interactive behaviors via edge topology reasoning compared with the human annotations of interactive agents (rendered in red). Moreover, BeTopNet may source on potential interactions as rendered in grey.

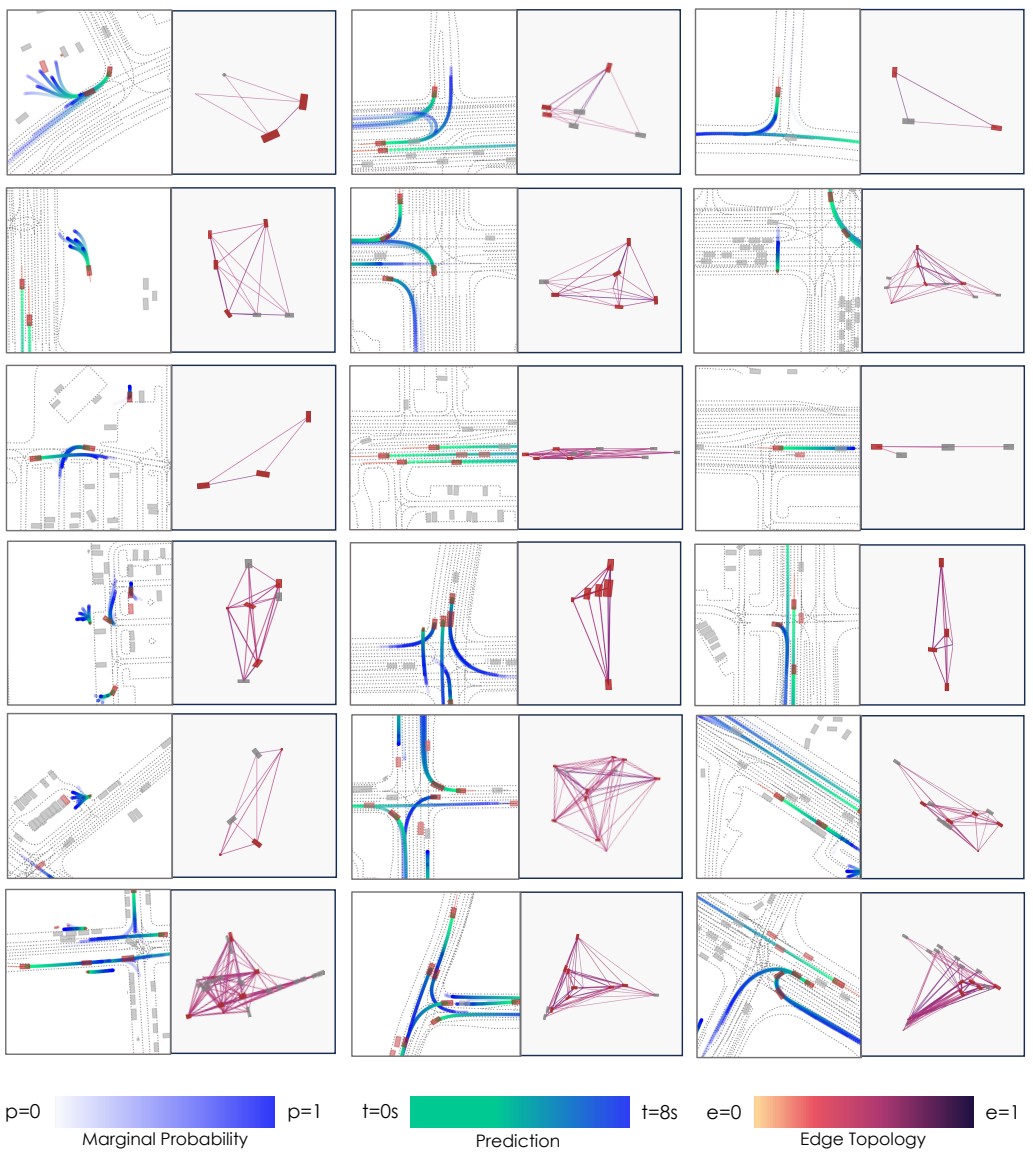

Figure 10: **Qualitative results of BeTopNet in WOMD marginal prediction.** BeTopNet performs compliant and accurate marginal predictions on multiple agents, reasoning diverse edge topology which stabilizes the behavioral patterns for future interactions.

