# OpenReview forum: "Reasoning Multi-Agent Behavioral Topology for Interactive Autonomous Driving"
_NeurIPS.cc/2024/Conference — NeurIPS 2024 poster_

### Official Review · Reviewer_aQzs · 2024-07-13

**Soundness:** 3
**Presentation:** 3
**Contribution:** 3
**Rating:** 7
**Confidence:** 3

**Summary:**

The paper presents a approach to address the challenges of autonomous driving in multi-agent scenarios and heterogeneous interaction. It introduces the concept of Behavioral Topology (BeTop) and its corresponding network BeTopNet. BeTop is based on braid theory and aims to provide a topological representation of multi-agent interactions to enhance the prediction and planning of autonomous vehicles (AVs). The proposed method focuses on creating a compliant behavioral pattern among agents, which guides the trajectory generation for AVs. Extensive experiments on large-scale datasets such as nuPlan and WOMD demonstrate the good performance of BeTop in both prediction and planning tasks.

**Strengths:**

* The paper is well-written and polished.
* The idea of using braid theory to explicit formulate the interactions among agents is interesting and lays a solid mathematical foundation for BeTop.
* The proposed method effectively integrates prediction and planning in a unified framework.
* The proposed method and baselines are extensively evaluated on two large-scale real-world datasets to demonstrate the performance on both motion prediction and planning.
* Ablation studies show that each component of the proposed method contribute to its performance on the planning task.

**Weaknesses:**

* The proposed method infer one type of agent behavior topology from one mode of future trajectories (e.g., 8s), while the topology of multi-agent interaction for real-world autonomous driving is usually multi-modal and dynamic. A discussion of modeling multi-step and dynamic topologies of vehicles for a long horizon would be beneficial.

**Questions:**

* The proposed method infer one type of agent behavior topology from one mode of future trajectories (e.g., 8s), while the topology of multi-agent interaction for real-world autonomous driving is usually multi-modal and dynamic. A discussion of modeling multi-step and dynamic topologies of vehicles for a long horizon would be beneficial.

**Limitations:**

* The current implementation of BeTop considers only one-step future topology. Extending this to multi-step reasoning could provide more robust predictions and planning.

---

> ### Author Rebuttal · Authors · 2024-08-06
>
> Thanks for your valuable comments. We address your questions below.
>
> >**W1/Q1:** The proposed method infer one type of agent behavior topology from one mode of future trajectories (e.g., 8s), while the topology of multi-agent interaction for real-world autonomous driving is usually multi-modal and dynamic. A discussion of modeling multi-step and dynamic topologies of vehicles for a long horizon would be beneficial.
>
>
> Thanks for your insightful suggestions. As mentioned in Lines 300-301, another challenge and direction is the development of Multi-step BeTop. In this paper, we clarify that multi-step BeTop refers to using multiple intervals to summarize entire future interactions with different temporal granularities.
>
> To explore this, we add a *new ablation experiment* to evaluate the effect of multi-step BeTop with minimal adjustments to the current BeTopNet framework. In our study, future interactions are split into 1 (base), 2, 4, and 8 steps (intervals) for multi-step BeTop labels. The multi-step topology reasoning task is then deployed through the current BeTopNet decoder, with an expanded MLP Topo Head for output steps. A max-pooling over BeTop steps is performed to comply with the indexing for topo-guided attention. The ablation results are as follows:
>
> |BeTop reasoning steps/intervals|mAP|minADE|minFDE|Miss Rate|Inference Latency (ms)|Training Latency (ms)| # Params. (M)|
> |-----------------------------------|-----------|-----------|-----------|-----------|------------------------|-----------------------|------------|
> |1/Base|0.392|0.637|1.328|0.144|70.0| 101.6|45.380|
> |2|**0.394**|**0.633**|**1.325**|0.145|75.5|110.6|45.382|
> |4|0.391|0.634|1.326|**0.142**|80.0|133.4|45.386|
> |8|0.389|0.641|1.347|0.147|90.0|255.0|45.393|
>
> Compared to the baseline 1-step reasoning, multi-step BeTop reasoning slightly improves BeTopNet's performance (e.g., 2-steps, +0.2 mAP), with a corresponding increase in computational costs for additional steps.
>
> This result highlights the potential of multi-step reasoning to enhance BeTopNet in interactive scenarios. One-step BeTop performs relatively well because the current topo-guided attention is optimized for single-interval reasoning. However, the 8-step configuration shows a slight drop, which might result from the minimal adjustments for BeTopNet. Direct reasoning and Max-pool over multi-step BeTop at the topo-attention may not predict and capture multi-interval interactions effectively, leading to potential noise or information loss.
>
> The current approach focuses primarily on formulating and integrating BeTop into the integrated prediction and planning (IPP) tasks, but refining temporal granularity for more accurate and efficient interactions remains an open question. We believe how to effectively leverage multi-step BeTop represents an interesting area for future exploration. We will enrich the experiment and discussion above in the revision accordingly.

---

> ### Comment · Reviewer_aQzs · 2024-08-12
>
> I thank the authors for the extra results and clarification and I would raise my score.

---

> > ### Author Response · Authors · 2024-08-13
> > **Response to the Reviewer**
> >
> > Thanks for the feedback and raising the score! We do appreciate your helpful review and will update the paper accordingly.

---

### Official Review · Reviewer_PsGi · 2024-07-13

**Soundness:** 4
**Presentation:** 3
**Contribution:** 4
**Rating:** 7
**Confidence:** 4

**Summary:**

The paper introduces a novel approach to enhance the safety and social consistency of autonomous driving systems through improved multi-agent behavioral integration. To address inefficiencies and inconsistencies in current behavioral representations, the authors propose Behavioral Topology (BeTop), a topological framework derived from braid theory that captures consensual behavioral patterns among multiple agents. This framework guides downstream trajectory generations and ensures stable collective behavior when integrating prediction and planning. The paper also presents BeTopNet, a synergistic learning framework supervised by BeTop that manages behavioral uncertainty and enhances prediction and planning consistency. Extensive experiments on large-scale real-world datasets, including nuPlan and WOMD, demonstrate that BeTop achieves state-of-the-art performance in prediction and planning tasks, showcasing its effectiveness in interactive scenarios

**Strengths:**

The paper introduces a novel approach to enhance the safety and social consistency of autonomous driving systems through improved multi-agent behavioral integration. To address inefficiencies and inconsistencies in current behavioral representations, the authors propose Behavioral Topology (BeTop), a topological framework derived from braid theory that captures consensual behavioral patterns among multiple agents. This framework guides downstream trajectory generations and ensures stable collective behavior when integrating prediction and planning. The paper also presents BeTopNet, a synergistic learning framework supervised by BeTop that manages behavioral uncertainty and enhances prediction and planning consistency. Extensive experiments on large-scale real-world datasets, including nuPlan and WOMD, demonstrate that BeTop achieves state-of-the-art performance in prediction and planning tasks, showcasing its effectiveness in interactive scenarios

**Weaknesses:**

In general, the paper is well written and there are no major weakness, however, there are some aspects that can be further discussed in the paper: 1.While the paper demonstrates effectiveness on specific datasets, it remains uncertain how well the method generalizes to diverse driving environments and conditions not covered in the training data. 2. The computational overhead associated with the topological framework and synergistic learning might be higher compared to simpler models, possibly affecting real-time performance.

**Questions:**

1. It is unclear how is BeTop used during inference when future trajectories for surrounding agents are unavailable.

2. It is unclear how are $Q_R$ and $Q_A$ initialized and defined.

3. How do you ensure the robustness of BeTopNet in highly dynamic and unpredictable driving environments, such as those with sudden changes or unexpected behaviors from other agents?

**Limitations:**

Please refer to the weakness section.

---

> ### Author Rebuttal · Authors · 2024-08-06
>
> Thank you for appreciating our work. We address your questions below.
>
> > **W1:** While the paper demonstrates effectiveness on specific datasets, it remains uncertain how well the method generalizes to diverse driving environments and conditions not covered in the training data.
>
> Thank you for your question. As mentioned in Lines 233-234 and Lines 239-240, we evaluate the performance of our method on both prediction and planning tasks using independent test sets, specifically Test14 and WOMD Test, as detailed in Tables 2-5 of the main text. These test sets are all private testing scenarios against the coverage in training data, ensuring that our method is evaluated objectively.
>
> Furthermore, as noted in Lines 708-712, the validation sets in nuPlan may share certain scenarios with the training set. Therefore we argue that the widely-used Val14 benchmark may not be fully representative, and have thus placed the Val14 results in the appendix (Table 9). In the main text, we focus on reporting test results for a more objective assessment of our method's generalizability.
>
> > **W2:** The computational overhead associated with the topological framework and synergistic learning might be higher compared to simpler models, possibly affecting real-time performance.
>
> Agreed. This may potentially lie in computations with model scale and decoded agents, which are the common challenges for the learning-based predictors with multi-agent settings. A preliminary computation study is conducted for BeTopNet against the MTR baseline using a single A100 GPU:
>
> | Method| Latency (ms) | GPU Memory (G) |
> |-|-|-|
> | MTR [22]| 84|5.2|
> | BeTopNet (Ours) | 89|  6.5 |
>
> We can observe that BeTopNet reports comparable latency and memory costs compared to MTR, with better prediction accuracy shown in the paper. The similar latency is due to the topo-guided attention, which reduces the KV features in agent aggregation during decoding, thereby decreasing the main computation cost for multi-head attention tensors. While BeTop introduces extra computations for reasoning, it requires more GPU memory for cached topology tensors. In practice, these computational challenges might be addressed through knowledge distillation or half-precision computations to reduce GPU requirements. We will enrich the above discussion in our revised version.
>
>
> > **Q1:** It is unclear how is BeTop used during inference when future trajectories for surrounding agents are unavailable.
>
> BeTop serves as a label to supervise the reasoning process in BeTopNet. During both training and inference, the reasoned BeTop is used within BeTopNet to guide predictions. This can be referred to Figure 3, Reason heads in Lines 196-197, and Training loss in Lines 205-206. We will improve the content accordingly for better understanding.
>
> > **Q2:** It is unclear how are $𝑄_𝑅$ and $𝑄_𝐴$ initialized and defined.
>
> $\mathbf{Q}_A$ is initialized by learnable embeddings. For the prediction task, the embedding will also be added with anchored features. These anchored features are predefined by K-means end-point anchors, referred to the clustering process in MTR [22].
>
> $\mathbf{Q}_R$ is initialized by MLP encoding of relative features of $\mathbf{S}_R$ [63].
>
> More details can be referred to Lines 637-639 in the Appendix and references [22, 63]. We will enrich extra clarifications in the revised main context accordingly.
>
> > **Q3:** How do you ensure the robustness of BeTopNet in highly dynamic and unpredictable driving environments, such as those with sudden changes or unexpected behaviors from other agents?
>
> Thank you for the insightful question. Our method has tried to tackle and validate these challenges as follows:
>
> - **Methodology**: BeTopNet ensures interactive robustness through its synergistic decoder design and topo-guided attention, which iteratively refines predictions by focusing on potential future interactions (by reasoned BeTop), and reacts to possible changes in the next layer of decoding.  For prediction/planning output, we use a Gaussian Mixture Model (GMM) to account for dynamic uncertainty. The contingency learning paradigm formulates predictions and planning guidance that balance short-term safety and long-term compliance. Expressely, this paradigm enhances short-term (0-3s) safe planning by considering worst-case predictions and conducting long-term branching (3-8s) for various prediction uncertainty.
> - **Experiments**: BeTopNet's robustness can be demonstrated in Test14-Hard and Test14-Inter benchmarks. Test14-Hard highlights scenarios where rule-based planning agents perform poorly, indicating difficult environments, while Test14-Inter focuses on highly dynamic situations where physics-based planners struggle. Our strong results in these benchmarks verify BeTopNet's capability to tackle such challenging scenarios.

---

> > ### Comment · Reviewer_PsGi · 2024-08-12
> >
> > Thank you for the clarification and additional information. I will remain my score.

---

> > > ### Author Response · Authors · 2024-08-13
> > > **Response to the Reviewer**
> > >
> > > Thank you for the response and recognition! We appreciate your valuable review for improving our work.

---

### Official Review · Reviewer_M7XC · 2024-07-17

**Soundness:** 2
**Presentation:** 2
**Contribution:** 3
**Rating:** 6
**Confidence:** 3

**Summary:**

The paper addresses the challenges of autonomous driving by integrating behavior among interactive agents, specifically focusing on issues caused by multi-agent scene uncertainty and heterogeneous interactions.  To tackle this, the paper introduces a topological formation called Behavioral Topology (BeTop), derived from braid theory, to represent consensual behavioral patterns among multiple agents. This formulation guides downstream trajectory generations and enhances the consistency of behavior prediction and planning.

**Strengths:**

1.	The experimental results seem to support the authors' claims.
2.	It is developed based on exiting braids topology. Novel method for an existing problem setup.
3.	The use of braid theory to distill compliant interactive topology from multi-agent future trajectories seems a good and intuitive idea to me.

**Weaknesses:**

The paper is generally not well-written, with extensive use of ChatGPT leading to paragraphs that are hard to follow.

- As far as I understand, the paper uses braid topology for just one step in planning and prediction. In long-horizon planning and prediction, this may not be sufficient as the motion of vehicles is  no longer independent. Could you provide some basis to the idea why one step topology will be enough?

-  There has been prior work using braid topology for planning (https://ieeexplore.ieee.org/stamp/stamp.jsp?tp=&arnumber=9812118). Benchmarking your method against this prior work would provide a clearer picture of this method.

- The method appears very similar to existing methods like Wayformer, with the main difference being the use of braid topology. I would like to see a detailed comparison showing how much the encoding of braid topology improves performance compared to Wayformer, especially given that only one-step braid topology is used instead of long-horizon topology.

Overall, the paper has potential, and I would be happy to discuss more with you. I would be willing to increase my score if my questions and concerns are addressed satisfactorily.

**Questions:**

See weaknesses

---

> ### Author Rebuttal · Authors · 2024-08-06
>
> Thanks for your valuable suggestions and we really appreciate your comments. We have carefully revised the draft for better readibility. We address each of the questions and confusion on the weaknesses as follows.
>
> >**W1:** As far as I understand, the paper uses braid topology for just one step. Could you provide some basis to the idea why one step topology will be enough?
>
> Thank you for the question. We would like to clarify that "one step" in the BeTop label refers to an interval summarizing interactions across the entire future horizon (T=8s, 10Hz), rather than a single timestep, illustrated in Lines 143-145. As detailed in Line 144, $e_{ij}=1$ signifies any future interaction between agents $i$ and $j$, while $e_{ij}=0$ indicates no interaction across all timesteps. Variations in BeTop labeling capture interactions over the full horizon with different temporal granularity.
>
> As mentioned in Lines 300-301, another challenge and direction for scalability is the development of Multi-step BeTop. In this paper, we clarify that multi-step BeTop refers to using multiple intervals to summarize entire future interactions with different temporal granularities.
>
> **We refer to the multi-step results and discussions in the Global Rebuttal above.**
>
> >**W2:** There has been prior work using braid topology for planning (https://ieeexplore.ieee.org/stamp/stamp.jsp?tp=&arnumber=9812118). Benchmark would provide a clearer picture of this method.
>
> Thank you for your question. The work in reference [43] has been cited in the paper, but we understand that it may cause some confusion. The differences of our work from [43] are as follows:
>
> - The paper [43] *does not involve planning tasks*, but rather focuses on traffic scenario analysis using topological braids to calculate an Interaction Score, as in Lines 88-90. In contrast, BeTop formulates behavioral topology and integrates it into joint prediction and planning using BeTopNet.
> - In terms of formulation,  [43] converts braid interactions into braid words, which is suitable for calculating the TC index but challenging for reasoning due to exponential complexity. While BeTop uses topological representations to simplify the formulation process, allows for reasoning and planning tasks with quadratic complexity.
> - While [43] conducts case studies on real-world datasets using log-replayed trajectories, BeTopNet includes reasoning in both prediction and planning benchmarks. This distinction highlights how BeTop goes beyond simple quantification to actively engage in prediction and planning processes.
>
> We will enrich these discussions to provide further clarity.
>
> >**W3:** I would like to see a detailed comparison showing how much the encoding of braid topology improves performance compared to Wayformer, especially given that only one-step braid topology is used instead of long-horizon topology.
>
> Thank you for your question. We understand that the similarities in the encoder-decoder structure might cause some confusion, so we we'd like to firstly highlight a few key differences between BeTopNet and Wayformer [61]. In the meantime, we conduct *additional ablation studies* to demonstrate BeTop's effectiveness in the Wayformer framework.
>
> **Key Differences:**
>   - **Model Foundations**: As mentioned in Section 3.2 in the main paper and Section C.1 in the Appendix, BeTopNet follows the encoder and anchor-based decoder design of Motion Transformer (MTR) [22]. While Wayformer focuses on encoder novelty, BeTopNet centers on decoder design, employing a synergistic Transformer decoder that jointly decodes trajectories and reasons BeTop interactions. This is contributed by topo-guided attention, which leverages reasoned topology for compliant predictions.
>   - **Encoding**: BeTopNet's encoder is based on MTR and uses local self-attention to encode all scene agents. Wayformer handles surrounded agents to avoid out-of-memory issues.
>   - **Decoding Strategies**: BeTopNet uses iterative refining strategies to selectively aggregate encoded features and decode trajectories for each decoder layer. In contrast, Wayformer uses stacked vanilla Transformer decoders for one-shot trajectory output.
>   - **Attention Design**: Wayformer’s latent query attention focuses on encoding query reduction, similar to MTR's local attention. BeTopNet uses guided attention for cross-attention in decoder, with reduction based on sorted BeTop results supervised by BeTop labels.
>
> Given differences, our studies primarily compare BeTopNet against MTR . In the rebuttal, we conduct additional ablations with BeTop in Wayformer to understand the generalization under different model foundations.
>
> **Ablative Experiments:**
>   - **Wayformer+BeTop**: The impact of using BeTop as additional supervision for interactions within the vanilla Wayformer model.
>   - **Wayformer+BeTopNet**: This combines the Wayformer encoder with the BeTopNet decoder design, showcasing our key contribution of topo-guided attention and iterative reasoning for complex interactions.
>
> The ablations use protocols outlined in Lines 274-277 with end-to-end decoding approach (Modality=6). We use the reproduction from (https://github.com/vita-epfl/UniTraj), since Wayformer is not open-sourced. The results are as follows:
>
> |Method|mAP|minADE|minFDE|Miss Rate|
> |-|-|-|-|-|
> |Wayformer|0.281|0.661|1.417|0.199|
> |Wayformer+BeTop|0.290|0.637|1.364|0.178|
> |**Wayformer+BeTopNet**|**0.344**|**0.604**|**1.261**|**0.166**|
>
>  Compared to the vanilla Wayformer, incorporating BeTop as supervision improves performance with a -6.2% Miss Rate and +3.2% mAP.  Furthermore, integrating BeTopNet significantly boosts performance, achieving a +18.6% mAP and -7.2% Miss Rate. This enhancement is due to our ***synergistic decoder***, which uses iterative BeTop reasoning and Topo-guided attention to refine trajectories by selectively aggregating interactive features. These results demonstrate BeTopNet's superior ability to enhance prediction and planning.

---

> > ### Comment · Reviewer_M7XC · 2024-08-12
> > **Thank for your rebuttal.**
> >
> > Hi Authors,
> >
> > Thank you for the clarification and doing experiments within this short time. I am increasing my score. All the best!

---

> > > ### Author Response · Authors · 2024-08-13
> > > **Response to the Reviewer**
> > >
> > > Thanks for your feedback and raising the score! We will integrate your insightful comments in our revision accordingly.

---

### Official Review · Reviewer_PWLC · 2024-07-20

**Soundness:** 3
**Presentation:** 3
**Contribution:** 3
**Rating:** 5
**Confidence:** 2

**Summary:**

This paper introduces a new approach, called Behavioral Topology (BeTop), to address the challenges in modeling multi-agent behaviors in autonomous driving. By utilizing braid theory, BeTop explicitly represents the consensual behavioral patterns among multiple agents, facilitating better prediction and planning. The framework, BeTopNet, incorporates this topological reasoning into a synergistic learning model that guides both behavioral prediction and planning. Good experiments on large-scale datasets, such as nuPlan and WOMD, demonstrate the superior performance of BeTopNet in both prediction and planning tasks, showcasing significant improvements over existing methods.

**Strengths:**

Good Presentation: The paper is well-organized and clearly presents the motivation, methodology, and results. The introduction of BeTop is logically structured, and the figures help in understanding the complex concepts.

Reasonable Formulation: The use of braid theory to represent multi-agent interactions is innovative and provides a solid theoretical foundation. This formulation helps in capturing the interactive behaviors more effectively compared to traditional dense or sparse representations.

Extensive Experiments: The authors have conducted comprehensive experiments on large-scale real-world datasets. These experiments cover both prediction and planning tasks, providing a thorough evaluation of the proposed method.

Performance Improvement: The experimental results demonstrate that BeTopNet achieves improved performance in prediction and planning tasks, especially in planning scores and prediction accuracy, with detailed metrics provided to back these claims.

**Weaknesses:**

Lack of Discussion on Multi-Agent Settings: While the paper introduces a topological approach for multi-agent behavior modeling, it lacks an in-depth discussion on how this method scales and handles various multi-agent settings. More insights into the limitations and potential scalability issues would strengthen the paper.

Formulation for Multi-Agent Settings: The paper could benefit from a more detailed formulation of the multi-agent setting. While the braid theory is used to model interactions, a clearer and more comprehensive explanation of how this integrates with different numbers and types of agents would be helpful.

**Questions:**

Lack of Discussion on Recursive Settings: While the paper introduces a topological approach for multi-agent behavior modeling, it lacks an in-depth discussion on how this method scales and handles various steps of multi-agent settings. More insights into the limitations and potential scalability issues would strengthen the paper. While the braid theory is used to model interactions, a clearer and more comprehensive explanation of how this integrates with different numbers and types of agents would be helpful.

**Limitations:**

Yes.

---

> ### Author Rebuttal · Authors · 2024-08-06
>
> Thanks for your appreciation and the helpful review of our work. We address your concerns below.
>
> >**W1/Q1:** Lack of Discussion on Multi-Agent Settings: While the paper introduces a topological approach for multi-agent behavior modeling, it lacks an in-depth discussion on how this method scales and handles various steps of multi-agent settings. More insights into the limitations and potential scalability issues would strengthen the paper.
>
>
> 1. **How does this method scale:**
>
> - **Scaling model size:** In the rebuttal, we *additionally* ablate BeTopNet with different model scales. We adjust the number of decoding modalities and feature dimensions to get three sizes of models. The results are as follows and we have added the experiment in the revision.
>
> | Method| Description|mAP|Miss Rate|Latency (ms)|# Params. (M)|
> |-|-|-|-|-|-|
> | BeTopNet-Small|Mode=6, DecDim=256|0.391|0.131| 45|28.91|
> | BeTopNet-Medium|Mode=64, DecDim=256|0.437|0.119|65|28.91|
> | BeTopNet-Base|Mode=64, DecDim=512|**0.442**|**0.117** |70|45.38|
>
> - **Scaling number of decoded agents:** Besides, we provide the computational complexity when the number of decoded agents increase during prediction, which is also a common challenge for the multi-agent setting. A preliminary computation study is conducted for BeTopNet against MTR:
>
> |Method|# Decode Agents|Latency (ms)|GPU Memory (G)|
> |-|-|-|-|
> ||
> | MTR [22]|8|84|5.2|
> | MTR [22]|16|123|7.1|
> | MTR [22]|32|193|15.6|
> ||
> |BeTopNet (Ours)|8|89|6.5|
> |BeTopNet (Ours)|16|120|10.8|
> |BeTopNet (Ours)|32|166|19.2|
>
> We can observe that BeTopNet reports comparable latency and memory costs compared to MTR, with better prediction accuracy shown in the paper. The similar latency is due to the topo-guided attention, which reduces the KV features in agent aggregation during decoding, thereby decreasing the main computation cost for multi-head attention tensors. While BeTop introduces extra computations for reasoning, it requires more GPU memory for cached topology tensors. In practice, these computational challenges might be addressed through knowledge distillation or half-precision computations to reduce GPU requirements. We will enrich the above discussion in our revised version.
>
> 2. **How does this method handle various steps of multi-agent setting:** As mentioned in Lines 300-301, another challenge and direction for scalability is the development of Multi-step BeTop. In this paper, we clarify that multi-step BeTop refers to using multiple intervals to summarize entire future interactions with different temporal granularities.
>
> To explore this, we add a *new ablation experiment* to evaluate the effect of multi-step BeTop with minimal adjustments to the current BeTopNet framework.
>
> **We refer to the Multi-step results and discussions in the *Global Rebuttal* above.**
>
>
>
> >**W2:** Formulation for Multi-Agent Settings: The paper could benefit from a more detailed formulation of the multi-agent setting. While the braid theory is used to model interactions, a clearer and more comprehensive explanation of how this integrates with different numbers and types of agents would be helpful.
>
> Thanks for your question.
>
> 1. To have a more intuitive understanding of how BeTop deals with varied agent numbers and categories:
>
>   - For **varied agent numbers**:  Each BeTop would outline whole scene agents. We leverage batched padding by the maximum scene agent number per batch, and generate the padding mask during batched BeTop formulation. Hence, BeTop can be formulated uniformly by batched padding mask.
>   - For **varied agent types**: Variations in agent types would not affect the formulation process. BeTop is only defined through states $(x, y, \theta)$ from multi-agent future trajectories. In fact, the agent types are considered in multi-agent states $\mathbf{X}$ and queries $\mathbf{Q}_A$ as model inputs.
>
> 2. The detailed formulation of BeTop (Lines 127-146) is summarized for the multi-agent future interactions.
>
> **The inputs :**
>
>   - (1) Multi-agent future trajectories $\mathbf{Y}$ of states $(x, y, \theta)$; Tensor shape: $[N_a, T_f, 3]$. (We omit subscript $a$ and $f$ below for simplicity.)
>   - (2) Agent padding mask $M$; Tensor shape: $[N]$
>
> **The formulation process:**
>
> Loop for index $i$, $j$ from 1 to $N$;
>
>   - (1) Calculate $e_{ij}$ by firstly index the future state tensor of sourced agent $\mathbf{Y}[i]$ and targeted agent $\mathbf{Y}[j]$; Both tensors have tensor shape $[T, 3]$;
>   - (2) Conduct the LocalTransformation() function by agent $i$ for sourced agent $\mathbf{Y}[i]$ and targeted agent $\mathbf{Y}[j]$. This is linked to the Braid function mapping by $(f^i_i, f^i_j)$ in Line 132-134, and refers to the process in Line 143-144; Both tensors output shape $[T]$ for lateral states;
>   - (3) Passed through the `SegmentIntersect()` function [76] for locally transformed $\mathbf{Y}[i]$ and $\mathbf{Y}[j]$ referred to the function `I(.,.)` in Line 144 with shape $[T]$; Conduct the max-pooling across $T$ for the finalized $e_{ij}$ summarizing the future interactions as one-step (one-interval)
>   - (4) Let $\mathcal{E}\_{ij} = e\_{ij}$
>
> Expand the agent padding mask $MM^T$ ( tensor shape: $[N, N]$), and mask BeTop $\mathcal{E}$
>
> **Output:** masked BeTop $\mathcal{E}$ for varied agents; Tensor shape: $[N, N]$
>
> In practice, the formulation process is conducted in batches and calculated using parallelized tensor multiplications instead of the loop iterations for more efficient computation.
>
> We hope the detailed formulations will aid in a better understanding of the multi-agent settings.

---

> ### Comment · Reviewer_PWLC · 2024-08-14
> **Response**
>
> Thanks for the detailed response. I do not have other questions, though some of my concerns remain. But in general, I see fit to accept this work.

---

> > ### Author Response · Authors · 2024-08-14
> > **Response to the Reviewer**
> >
> > Thank you for the kind feedback. Your time and effort in reviewing our work are truly appreciated! We will revise the manuscript according to your valuable comments. During the remaining author-reviewer discussion period, we would be glad to provide further clarifications for any concerns you may have.

---

### Author Rebuttal · Authors · 2024-08-06

*Dear Area Chairs and Reviewers,*

*We thank all the Reviewers for their careful reviews and valuable comments on our work. We have taken each comment into consideration, added more ablative experiments in the rebuttal, and clarified some technical details. Please see each response below. We are grateful for the opportunity to improve our work with your guidance.*

*Best Regards,*
*The Authors*


**Here we refer to the general question proposed by #Rew.PWLC and #Rew.M7XC:**

**Ablations and discussions for multi-step:** As mentioned in Lines 300-301, another challenge and direction for scalability is the development of Multi-step BeTop. In this paper, we clarify that multi-step BeTop refers to using multiple intervals to summarize entire future interactions with different temporal granularities.

To explore this, we add a *new ablation experiment* to evaluate the effect of multi-step BeTop with minimal adjustments to the current BeTopNet framework. In our study, future interactions are split into 1 (base), 2, 4, and 8 steps (intervals) for multi-step BeTop labels. The multi-step topology reasoning task is then deployed through the current BeTopNet decoder, with an expanded MLP Topo Head for output steps. A max-pooling over BeTop steps is performed to comply with the indexing for topo-guided attention following the current design in Line 144. The ablation results are as follows:

|BeTop reasoning steps/intervals|mAP|minADE|minFDE|Miss Rate|Inference Latency (ms)|Training Latency (ms)| # Params. (M)|
|-|-|-|-|-|-|-|-|
|1/Base|0.392|0.637|1.328|0.144|70.0| 101.6|45.380|
|2|**0.394**|**0.633**|**1.325**|0.145|75.5|110.6|45.382|
|4|0.391|0.634|1.326|**0.142**|80.0|133.4|45.386|
|8|0.389|0.641|1.347|0.147|90.0|255.0|45.393|

Compared to the baseline 1-step reasoning, multi-step BeTop reasoning slightly improves BeTopNet's performance (e.g., 2-steps, +0.2 mAP), with a corresponding increase in computational costs for additional steps.

This result highlights the potential of multi-step reasoning to enhance BeTopNet in interactive scenarios. One-step BeTop performs relatively well because the current topo-guided attention is optimized for single-interval reasoning. However, the 8-step configuration shows a slight drop, which might result from the minimal adjustments for BeTopNet. Direct reasoning and Max-pool over multi-step BeTop at the topo-attention may not predict and capture multi-interval interactions effectively, leading to potential noise or information loss.

The current approach focuses primarily on formulating and integrating BeTop into the integrated prediction and planning (IPP) tasks, but refining temporal granularity for more accurate and efficient interactions remains an open question. We believe how to effectively leverage multi-step BeTop represents an interesting area for future exploration. We will enrich the experiment and discussion above in the revision.

*Please refer to the rebuttal modules below for our point-to-point responses to each reviewer.*

---

### Decision · Program_Chairs · 2024-09-25

**Decision:**

Accept (poster)

**Comment:**

This paper introduces Behavioral Topology (BeTop), a novel approach to modeling multi-agent behaviors in autonomous driving using braid theory. The authors present BeTopNet, a framework that incorporates topological reasoning into a synergistic learning model for both behavioral prediction and planning.

The reviewers generally agree on the innovative nature of the approach. Reviewer PWLC notes that "the use of braid theory to represent multi-agent interactions is innovative and provides a solid theoretical foundation," while Reviewer aQzs states that it "lays a solid mathematical foundation for BeTop." The comprehensive evaluation is another strong point, with Reviewer PWLC highlighting the "extensive experiments on large-scale real-world datasets" and Reviewer PsGi confirming that these experiments "demonstrate that BeTop achieves state-of-the-art performance in prediction and planning tasks."

However, the paper has some limitations. Reviewer PWLC points out that it "lacks an in-depth discussion on how this method scales and handles various multi-agent settings." A major concern raised by multiple reviewers (M7XC, PsGi, and aQzs) is the use of only one-step topology. Reviewer M7XC questions, "Could you provide some basis to the idea why one step topology will be enough?" Reviewer M7XC also suggests benchmarking against prior work using braid topology for planning to provide a clearer picture of the method's contribution.

The presentation of the paper received mixed reviews. While most reviewers found it well-organized, Reviewer M7XC notes that "The paper is generally not well-written, with extensive use of ChatGPT leading to paragraphs that are hard to follow."

Several important questions were raised by the reviewers. Reviewer PsGi asks how BeTop is used during inference when future trajectories for surrounding agents are unavailable, and how the robustness of BeTopNet is ensured in highly dynamic and unpredictable driving environments. Reviewer PWLC suggests providing a clearer explanation of how the method integrates with different numbers and types of agents.

Recommendation:
Despite the concerns raised, the paper's innovative approach and comprehensive experiments are seen as significant contributions to the field. Three out of four reviewers recommend acceptance (two "Accept" and one "Weak Accept"), while one reviewer gives a "Borderline Accept."

Based on these reviews, the recommendation is to accept the paper, contingent on the authors addressing the following key points in the camera-ready version:

Provide a more detailed discussion on multi-step topologies and their potential benefits.
1. Clarify how the method scales to various multi-agent settings.
2. Include a comparison with existing methods, particularly those using braid topology for planning.
3. Address the questions raised about inference and robustness in dynamic environments.
4. Improve the clarity and readability of the paper, addressing the writing concerns raised by Reviewer M7XC.